# METAMETRICS: CALIBRATING METRICS FOR GENERATION TASKS USING HUMAN PREFERENCES

**Genta Indra Winata**[1][†][*]**, David Anugraha**[2][†]**, Lucky Susanto**[3][†]**,**
**Garry Kuwanto**[4][†]**, Derry Tanti Wijaya**[3,4]
[1]Capital One   [2]University of Toronto   [3]Monash University Indonesia   [4]Boston University
genta.winata@capitalone.com, anugraha@cs.toronto.edu,
lucky.susanto@monash.edu, {gkuwanto,wijaya}@bu.edu

## ABSTRACT

Understanding the quality of a performance evaluation metric is crucial for ensuring that model outputs align with human preferences. However, it remains unclear how well each metric captures the diverse aspects of these preferences, as metrics often excel in one particular area but not across all dimensions. To address this, it is essential to systematically calibrate metrics to specific aspects of human preference, catering to the unique characteristics of each aspect. We introduce METAMETRICS, a calibrated meta-metric designed to evaluate generation tasks across different modalities in a supervised manner. METAMETRICS optimizes the combination of existing metrics to enhance their alignment with human preferences. Our metric demonstrates flexibility and effectiveness in both language and vision downstream tasks, showing significant benefits across various multilingual and multi-domain scenarios. METAMETRICS aligns closely with human preferences and is highly extendable and easily integrable into any application. This makes METAMETRICS a powerful tool for improving the evaluation of generation tasks, ensuring that metrics are more representative of human judgment across diverse contexts.

## 1 INTRODUCTION

Evaluating machine-generated sentences has long been one of the main challenges in natural language processing (NLP). Callison-Burch et al. (2006) provide multiple examples where a high BLEU score does not necessarily indicate true sentence similarity, and conversely, highly similar sentence pairs can receive low BLEU scores. BERTScore (Zhang et al., 2019) is designed to capture semantic similarities, also falls short in capturing the all the nuances to have a comprehensive evaluation. Figure 1 illustrates two such cases where traditional metrics either overestimate or underestimate the alignment of generated sentences. Such cases highlight a critical limitation for robust evaluation, as it may fail to account for the multifaceted aspects of language quality.

In light of recent advancements such as Reinforcement Learning with Human Feedback (RLHF) (Ouyang et al., 2022), ensuring that generated outputs align with human preferences has become increasingly critical. Models that optimize for human preference, rather than solely relying on traditional metrics, have demonstrated superior performance in producing content that aligns with human preference (Rafailov et al., 2024; Winata et al., 2024b). This shift highlights the need for evaluation metrics that accurately reflect human subjective judgments across multiple dimensions. Such metrics are essential for guiding models to generate more human-aligned outputs by comparing their results against human judgments. Metrics that exhibit a high correlation with human ratings are considered more reliable and effective for evaluating model performance.

This is particularly important for NLP tasks, where the subtleties of human language and context significantly influence quality assessments. For example, in machine translation (Freitag et al., 2023; Juraska et al., 2023), the accuracy and fluency of the translated text are critical factors considered by humans. Similarly, in text summarization (Fabbri et al., 2021), the coherence, relevance, and con-

---

[*]The work was done outside Capital One. [†]Equal contribution.

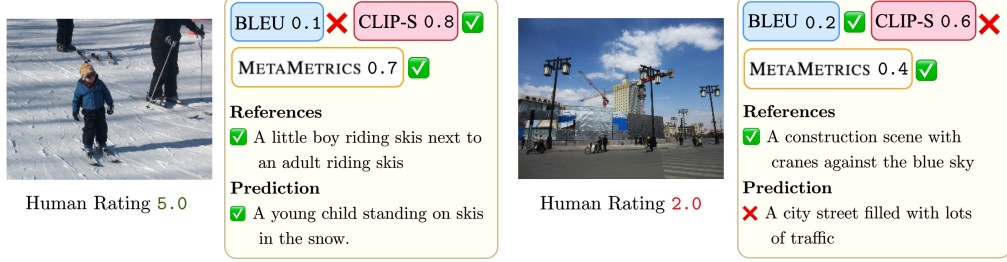

Figure 1: Examples of image captioning on THumB 1.0 dataset comparing METAMETRICS against BLEU and CLIP-S scores. METAMETRICS scores of predicted captions are closer to human ratings compared to BLEU and CLIP-S scores in the left and right images, respectively.

ciseness of the summary are key aspects that need evaluation. Beyond NLP tasks, vision-language (VL) tasks like image captioning also benefit from reliable evaluation metrics (Hessel et al., 2021). In these tasks, the generated captions must accurately describe the image content while maintaining grammatical correctness and contextual relevance. The complexity of these tasks highlights the need for metrics that can effectively capture the quality of the generated content in a manner consistent with human judgment. Each metric may be good for a specific human preference aspect, and it is necessary to identify which metric is suitable for each aspect to ensure comprehensive evaluation.

**Many Metrics Are Not Tuned on Human Preferences.** Standard metrics that are commonly used to evaluate downstream language tasks, such as BLEU (Papineni et al., 2002) and BERTScore (Zhang et al., 2019), often fail to align with human preferences. In many generation tasks, it is crucial to assess the quality of outputs based on human judgment, as these evaluations comprise multiple dimensions of quality. Therefore, we need a comprehensive framework that allows us to identify, refine, and utilize effective metrics for evaluating generated content in a way that aligns with human judges.

Motivated by this insight, we introduce METAMETRICS, a meta-metric designed to better align with human preferences by calibrating multiple metrics using human assessment scores. Our method is both efficient and fast, eliminating the need for extensive training. Our contributions are three-fold:

- We introduce METAMETRICS, a supervisedly calibrated, flexible, and modality- and language-agnostic metric that aligns closely with human preferences on **five different tasks**. Our metric is designed to operate in **two distinct settings**: reference-based and reference-free, providing versatility across a wide range of tasks and modalities.

- We demonstrate that METAMETRICS is adaptable to a wide range of requirements, including performance optimization and metric efficiency through a comprehensive benchmarking of diverse metrics across a range of language and vision tasks §4. We also show that METAMETRICS can be used effectively as a reward model.[1] This adaptability ensures that METAMETRICS can be calibrated to suit various human preference aspects.

- We demonstrate that METAMETRICS outperforms existing metrics by leveraging complementary metrics to enhance overall performance. We conduct a detailed analysis to understand how METAMETRICS attributes weight or attention to specific metrics, thereby measuring the contribution of each component. We will release the code and models to facilitate reproducibility and empower researchers and practitioners to utilize and extend our metric for various applications.[2]

## 2 AREN'T EXISTING METRICS GOOD ENOUGH?

The challenge of developing metrics tailored to specific tasks is not a new issue. For a long time, researchers have struggled to identify appropriate metrics that align with human preferences. In this

---

[1] METAMETRICS ranks 5th out of 155 on the overall leaderboard, utilizing the fewest parameters, and secures 3rd place on the detailed leaderboard at RewardBench. It reflects data as of November 16th, 2024.

[2] We release METAMETRICS as an open-source library at https://github.com/meta-metrics/metametrics.

section, we outline the rationale for the need for a new evaluation metric that is both suitable for our tasks and aligned with human preferences.

**Commonly Used Metrics are Not Robust and Unrepresentative.** Metrics such as Perplexity (PPL) (Jelinek et al., 1977; Bengio et al., 2000) and BLEU (Papineni et al., 2002) are widely used to measure the quality of generated text in various generation tasks. However, they are not always the most suitable metrics, particularly when accounting for variations in writing styles and minor character differences due to diacritics (Freitag et al., 2022). Critics have noted that PPL is an English-centric metric that performs well for English but is less effective for other languages, such as Japanese (Kuribayashi et al., 2021). THumB (Kasai et al., 2022) further revealed that widely used metrics fail to adequately capture the semantic quality and diversity of high-quality human annotations, leading to misleading evaluations of model performance, particularly for captions.

**Limited Capability of an Individual Metric.** Single metrics like BERTScore (Zhang et al., 2019), ROUGE (Lin, 2004), and METEOR (Banerjee & Lavie, 2005) are beneficial and capable of measuring the quality of generated content. However, they have significant limitations. For instance, in summarization tasks, BERTScore (Recall) excels in assessing consistency but falls short in evaluating coherence. Conversely, BERTScore (f1) performs well in measuring relevance but not consistency (Fabbri et al., 2021). This variability means that a single metric may perform well in one aspect but poorly in another, making it challenging to select the appropriate metric without extensive benchmarking. This ambiguity complicates the process of choosing and utilizing the most suitable metric for a specific use case.

**One Metric with Too Many Variants and Implementations.** Metrics like BERTScore allow the use of various BERT models, but the sheer number of options can make it difficult to identify the best one without a systematic approach. This can lead to an exhaustive search process. BERT models trained on English often underperform on non-English languages, and vice versa. Non-English languages in Latin script may also yield poorer results compared to those in their native scripts. Compounding these issues, many metrics are heavily parameterized, yet their settings are frequently undocumented, resulting in score variations across implementations (Post, 2018; Grusky, 2023).

To tackle these challenges, we propose a new paradigm for creating customizable metrics that are robust across various tasks and closely aligned with human preferences. Our approach will enable systematic evaluation and automatic selection of the most suitable combination of metrics that is aligned with human judgments on target tasks, ensuring optimal performance and relevance.

## 3 METAMETRICS

In this section, we outline the notations and definitions used in our proposed method, and detail the conditions required for optimizing the method based on human preferences. Additionally, we discuss the key factors influencing the optimization process.

### 3.1 PRELIMINARIES

**Definition.** We define $\theta_i$ as a metric function that maps a sample input $x$ to a score $\hat{y}_i$, where $i \in \{1, \ldots, N\}$ denotes different metrics. Each $\theta_i$ depends on the type of task it is applied to. For reference-based metric, the data is evaluated in the context of $x = (x_{\text{hyp}}, x_{\text{ref}})$, where $x_{\text{hyp}}$ and $x_{\text{ref}}$ correspond to the hypothesis text and the reference text, respectively. For reference-free metric, only $x_{\text{hyp}}$ will be used. In VL tasks, the input is extended to include an image, which would correspond to $x = (x_{\text{text}}, x_{\text{image}})$ where $x_{\text{text}}$ and $x_{\text{image}}$ correspond to the caption and image, respectively.

Given a set of $N$ evaluation metrics $\{\hat{y}_1, \ldots, \hat{y}_N\}$, we define $\Phi$ to compute a scalar meta-metric score of $\hat{y}_{\text{MM}}$. The idea of utilizing multiple metrics is to combine scores from multiple metrics regardless of the metric types. Overall, we define $\theta_{\text{MM}}$ as a meta-metric function where $\hat{y}_{\text{MM}}$ is computed as follows:

$$\hat{y}_i = \theta_i(x), \quad \hat{y}_{\text{MM}} = \theta_{\text{MM}}(x) = \Phi(\hat{y}_1, \cdots, \hat{y}_N). \tag{1}$$

The METAMETRICS $\theta_{\text{MM}}$ is used to calculate the objective value $\rho(\hat{y}_{\text{MM}}, z)$, where $\rho$ is measures alignment with $z \in \mathbb{R}$, the human preference score. In practice, $z$ can represent various scores annotated by human judges, such as coherence and fluency.

## 3.2 Human Preference Optimization

**Notations and Objective.** Recall that we aim to calibrate $\theta_{\text{MM}}$, that maximizes an objective calibration function $\rho(\hat{y}_{\text{MM}}, z)$, where $z$ denotes human assessment scores–encompassing any score annotated by human evaluators. $\rho$ is a function that measures alignment between $z$ and METAMETRICS scores, $\hat{y}_{\text{MM}}$. METAMETRICS is designed to combine scores of multiple metrics $\theta_1(x), \theta_2(x), \ldots, \theta_N(x)$, learning the weights $w_i$ to assign to each $\hat{y}_i = \theta_i(x)$ to maximize $\rho(\hat{y}_{\text{MM}}, z)$. Each metric has its score $\hat{y}_i$ ranges within a specific minimum and maximum value. Some metrics, particularly neural-based ones, can fall outside this defined range. To standardize these metrics, we need to normalize them to a common scale that ranges from 0 to 1. In this scale, 0 represents poor translation performance, while 1 represents perfect translation performance. We pre-process $\hat{y}_i$ before combining these scores during METAMETRICS training, as detailed in Appendix D.2. The advantage of METAMETRICS lies in its flexibility and adaptability to different tasks and domains. Certain metrics on certain tasks may exhibit strong correlations with human judgments, and by constructing a composite metric that learns to integrate these reliable metrics with human judgments, we can enhance the overall correlation with human evaluations.

## 3.3 Optimization Methods

In this work, we focus on two optimization methodologies to train METAMETRICS: Bayesian Optimization (BO) and Boosting. BO offers the advantage of interpretability, allowing us to clearly identify which metrics contribute most significantly to the final outcome. Conversely, Boosting excels in enhancing alignment and accounting for the compositionality of different metrics, even when dealing with more complex functions. Although we can measure the contribution of each metric, the clarity and distinctness of these contributions are more pronounced with BO compared to Boosting.

### 3.3.1 Bayesian Optimization (BO)

BO constructs a posterior distribution of functions, typically modeled as a Gaussian Process (GP), to represent the function being optimized. As observations accumulate, the posterior distribution becomes more precise, allowing the algorithm to identify which regions of the parameter space to explore and which to ignore. BO is particularly effective for predicting unknown functions based on observed data, making it a robust method for optimizing complex and costly-to-evaluate functions such as the correlation between evaluation metric scores and human assessment scores.

Formally, we want to solve for the optimum weights $\mathbf{w}^* = \arg\max_{\mathbf{w} \in \mathbf{W}} \rho(\hat{y}_{\text{MM}}(\mathbf{w}), z)$, where $\rho(\hat{y}_{\text{MM}}(\mathbf{w}), z)$ is the alignment between METAMETRICS score $\hat{y}_{\text{MM}}(\mathbf{w})$ with the human preference score $z$. In this paper, we apply linear weighting for our metrics. We define $\mathbf{w} = [w_1, w_2, \ldots, w_N]$, $\hat{y}_{\text{MM}}(\mathbf{w}) = \sum_{i \in N} w_i \hat{y}_i$. As part of our ablation study, we explore alternative weighting functions, such as multiplicative scoring, which calculates the score by applying a weighted sum to the product of two metric scores. However, we find that linear weighting outperforms the multiplicative approach. Further details can be found in the Appendix C and F.6.

We assume a GP prior: that the alignment between the weighted combination of metric scores and the human preference score follows a GP $\rho(\hat{y}_{\text{MM}}(\mathbf{w}), z) \sim \mathcal{GP}(\mu(\mathbf{w}), k(\mathbf{w}, \mathbf{w}'))$, where $\mu(\mathbf{w})$ is is the prior mean of the alignment and $k(\mathbf{w}, \mathbf{w}')$ is the kernel function that measures the similarity between two sets of weights $\mathbf{w}$ and $\mathbf{w}'$. After observing a set of weight vectors $\mathbf{W} = [\mathbf{w}_1, \mathbf{w}_2, \ldots, \mathbf{w}_k]$ and their corresponding alignments $\rho = [\rho_1, \rho_2, \ldots, \rho_k]$, the GP posterior can be updated to predict the alignment for a new weight vector $\mathbf{w}$. To guide the search for the optimal weights $\mathbf{w}$, BO maximizes a function and selects the next weight vector $\mathbf{w}_{k+1}$ to evaluate using the function. BO's function balances exploration and exploitation using the posterior mean and variance, iterating until convergence or budget exhaustion. While challenging, maximizing the function is computationally efficient with standard tools, making BO suitable for resource-intensive sampling scenarios. GP employs a multivariate Gaussian distribution, offering a sparse representation of metrics that enhances interpretability by identifying key contributors to the objective.

### 3.3.2 Boosting Method

The boosting method we investigate is Extreme Gradient Boosting (XGBoost) (Chen & Guestrin, 2016). Gradient-boosted trees have long been recognized as a robust technique, supported by extensive literature demonstrating their effectiveness (Friedman, 2001). To enhance the efficiency of our

metric, we implement iterative pruning to eliminate less important metrics from the input, resulting in a more compact and faster metric.

**Iterative-based Pruning.**    When training an XGBoost regressor model, we can initially use a large set of metrics. However, this approach is less efficient than the BO model at inference time. To address the scalability and efficiency issues, we propose an iterative pruning method (Algorithm 1). We conduct $k$ iterations of XGBoost training, starting from the full set of metrics and removing the metric with the least feature importance at each iteration. We return the best XGBoost model among $k$ iterations based on the cross-validation value of $\rho(\hat{y}_{MM}, z)$ measured during training.

## 4 EXPERIMENTS SETUP

In this work, we explore two optimization methodologies: BO and Boosting. BO provides interpretability, highlighting metrics that significantly impact the final outcome. In contrast, Boosting enhances alignment and addresses the compositionality of metrics, even for complex functions. While we can measure each metric's contribution through feature importance in Boosting, BO is more interpretable than Boosting. We also compare calibrating with all the metrics and calibrating only using top-5 correlated metrics from the tuning set. For each evaluation task in **abstractive text summarization**, **question answering**, and **image captioning**, we use a 30%-70% train-test split as no predefined split was available in the datasets used for these tasks. Alternatively, we follow existing standardized benchmarks for **machine translation** and **reward model scoring**.

**Abstractive Text Summarization.**    For this task, we use the SummEval (Fabbri et al., 2021) for text summarization evaluation. SummEval, based on CNN/DailyMail (Hermann et al., 2015), is rated by human annotators on coherence, consistency, fluency, and relevance using a 1 to 5 Likert scale, and we use the average human annotation score. For additional benchmarking, we also evaluate on BenchmarkLLM (BLLM) (Zhang et al., 2024) dataset, which includes both CNN/DailyMail and XSUM (Narayan et al., 2018) articles, with summaries assessed for faithfulness (binary scale), coherence, and consistency (1 to 5 Likert scale). For this dataset, the results are provided in the Appendix Table 10. We use the Kendall $\tau$ correlation function as our objective calibration function $\rho$. We refer to our metric as METAMETRICS-SUM (Table 1).

**Machine Translation.**    For this task, we train both reference-free and reference-based versions of the metric. We train our metric using the 3-year training data from WMT shared tasks datasets from 2020 to 2022 that are annotated using MQM annotation scores. We evaluate on MQM dataset from WMT23 and WMT24 shared task (Freitag et al., 2023; 2024). We use kendall $\tau$ correlation function as our objective calibration function $\rho$. We refer to our metric as METAMETRICS-MT (Table 2).

**Question Answering.**    For this task, we evaluate METAMETRICS performance across multiple QA subtasks, including Open Domain QA (Open-QA) (Rajpurkar, 2016; Berant et al., 2013; Petroni et al., 2019), Reading Comprehension QA (RCQA) (Bajaj et al., 2016; Fisch et al., 2019; Yang et al., 2018), and Reasoning QA. Open-QA retrieves answers from large, unstructured datasets like Wikipedia. RCQA requires the model to read a passage to derive answers directly from it, while Reasoning QA emphasizes logical inference, where answers cannot be directly extracted from the passage. Due to the nature of our paper, we require QA datasets with human evaluations. The data we use are collected from two sources: Evaluating Question Answering Evaluation (EQAE) (Chen et al., 2019) and EVOUNA (Wang et al., 2024a) and include Open-QA datasets: Natural Questions (NQ) (Kwiatkowski et al., 2019) and TriviaQA (TQ) (Joshi et al., 2017), RCQA dataset: NarrativeQA (Kočiský et al., 2018), and Reasoning QA dataset: SemEval 2018 Task 11 (SemEval) (Ostermann et al., 2018). We refer to our metric as METAMETRICS-QA (Table 3).

**Image Captioning.**    For this task, we use Flickr8k-Expert (Hodosh et al., 2013) and THumB 1.0 (Kasai et al., 2022) human evaluation datasets. Flickr8k-Expert consists 5,882 image-caption pairs, each having 3 human annotators with a scale of 1–4 where 1 is worst and 4 is best. The THumB 1.0 dataset is a rubric-based human evaluation framework applied to a subset of the MSCOCO dataset (Lin et al., 2014). The final score is of a scale of 1–5 where 1 is worst and 5 is best. We refer to our metric as METAMETRICS-CAP (Figure 2).

**Reward Model Scoring.**    We use METAMETRICS as reward model to score text generation from LLM. We use RewardBench (Lambert et al., 2024) as the test benchmark. We use cleaned Sky-

Table 1: Kendall and spearman correlation results with human ratings for summarization task on SummEval. For comprehensive results, please refer to Appendix Table 9. **Coh.**, **Cons.**, **Fluency**, and **Rel.** corresponds to coherence, consistency, and fluency, relevance, respectively. **Bold** and underlined values indicate the best and second best performance, respectively.

| Metric | Kendall | | | | | Spearman | | | | |
|---|---|---|---|---|---|---|---|---|---|---|
| | Coh. | Cons. | Fluency | Rel. | Avg. | Coh. | Cons. | Fluency | Rel. | Avg. |
| BLEU | 0.110 | 0.126 | 0.113 | 0.170 | 0.130 | 0.157 | 0.160 | 0.145 | 0.239 | 0.175 |
| chrF | 0.143 | 0.094 | 0.071 | 0.198 | 0.127 | 0.205 | 0.119 | 0.091 | 0.278 | 0.173 |
| METEOR | 0.077 | 0.102 | 0.072 | 0.162 | 0.103 | 0.108 | 0.130 | 0.093 | 0.229 | 0.140 |
| ROUGE-WE1 | 0.115 | 0.088 | 0.081 | 0.169 | 0.113 | 0.164 | 0.112 | 0.105 | 0.237 | 0.115 |
| BLEURT (Max) | 0.185 | 0.070 | 0.114 | 0.189 | 0.140 | 0.262 | 0.088 | 0.062 | 0.194 | 0.152 |
| BERTScore (f1) | 0.105 | 0.100 | 0.120 | 0.181 | 0.127 | 0.150 | 0.128 | 0.155 | 0.256 | 0.172 |
| LLM-BASED METRICS | | | | | | | | | | |
| BARTScore (Mean) | 0.086 | 0.074 | 0.040 | 0.143 | 0.086 | 0.123 | 0.094 | 0.052 | 0.202 | 0.118 |
| UniEval | 0.413 | 0.353 | 0.359 | 0.324 | 0.362 | 0.577 | 0.439 | 0.458 | 0.446 | 0.480 |
| G-Eval (GPT4) | 0.429 | 0.413 | 0.409 | 0.437 | 0.422 | 0.565 | 0.510 | 0.470 | 0.581 | 0.531 |
| ENSEMBLE BASELINES | | | | | | | | | | |
| Uniform | 0.141 | 0.133 | 0.117 | 0.213 | 0.151 | 0.201 | 0.170 | 0.150 | 0.298 | 0.205 |
| Weighted Avg | 0.150 | 0.141 | 0.122 | 0.220 | 0.159 | 0.215 | 0.179 | 0.158 | 0.309 | 0.215 |
| METAMETRICS-SUM | | | | | | | | | | |
| GP | 0.172 | 0.140 | 0.130 | 0.252 | 0.174 | 0.244 | 0.179 | 0.167 | 0.354 | 0.236 |
| XGBoost | 0.192 | 0.186 | 0.186 | 0.276 | 0.210 | 0.274 | 0.236 | 0.239 | 0.386 | 0.284 |
| w/ LLM-BASED METRICS | | | | | | | | | | |
| GP | 0.454 | 0.419 | 0.409 | **0.449** | 0.433 | 0.609 | 0.519 | 0.470 | **0.601** | 0.550 |
| GP (Top 2) | 0.461 | 0.428 | 0.409 | **0.449** | 0.437 | 0.628 | **0.528** | 0.470 | **0.601** | 0.557 |
| XGBoost | **0.476** | 0.367 | 0.404 | 0.447 | 0.424 | **0.642** | 0.460 | **0.512** | 0.600 | 0.553 |
| XGBoost (Top 2) | **0.476** | **0.436** | **0.430** | 0.445 | **0.447** | 0.636 | 0.508 | 0.511 | 0.594 | **0.562** |

Table 2: Kendall correlation results with human ratings on WMT23 (MQM). [†]The results are collected from Freitag et al. (2023). More detailed results for all metrics and their variants can be found in Table 12 in the Appendix. **Bold** and underlined values indicate the best and second best performance, respectively. Our METAMETRICS-MT correlates with human judgments better than or comparably to the best WMT evaluation metrics across different languages and evaluation settings.

| Metric | overall | en-de | | | he-en | | | zh-en | | |
|---|---|---|---|---|---|---|---|---|---|---|
| | sys/seg avg-corr | sys pearson | seg pearson | seg acc-t | sys pearson | seg pearson | seg acc-t | sys pearson | seg pearson | seg acc-t |
| REFERENCE-BASED METRIC | | | | | | | | | | |
| chrF[†] | 0.694 | 0.866 | 0.232 | 0.519 | 0.776 | 0.221 | 0.460 | 0.809 | 0.063 | 0.485 |
| BLEU[†] | 0.696 | 0.917 | 0.192 | 0.520 | 0.769 | 0.220 | 0.442 | 0.734 | 0.119 | 0.472 |
| BERTScore[†] | 0.742 | 0.891 | 0.325 | 0.528 | 0.895 | 0.335 | 0.515 | 0.810 | 0.236 | 0.499 |
| Yisi-1[†] | 0.754 | 0.925 | 0.366 | 0.542 | 0.917 | 0.395 | 0.529 | 0.823 | 0.290 | 0.504 |
| MetricX-23-XXL[†] | 0.808 | 0.977 | 0.585 | 0.603 | 0.910 | 0.548 | 0.577 | 0.873 | 0.625 | 0.531 |
| XCOMET-Ensemble[†] | 0.825 | 0.980 | **0.695** | 0.604 | 0.950 | 0.556 | 0.586 | 0.927 | **0.650** | 0.543 |
| COMET | 0.779 | 0.990 | 0.432 | 0.575 | 0.940 | 0.401 | 0.531 | 0.898 | 0.396 | 0.514 |
| ENSEMBLE BASELINES | | | | | | | | | | |
| Uniform | 0.824 | 0.989 | 0.688 | 0.610 | 0.954 | 0.546 | 0.588 | 0.935 | 0.641 | 0.543 |
| Weighted Avg | **0.827** | 0.989 | 0.690 | 0.613 | 0.955 | 0.545 | 0.587 | **0.937** | 0.642 | 0.545 |
| METAMETRICS-MT | | | | | | | | | | |
| GP | 0.819 | 0.970 | 0.638 | 0.610 | 0.947 | 0.546 | **0.590** | 0.900 | 0.646 | 0.539 |
| XGBoost | 0.825 | **0.992** | 0.680 | **0.616** | **0.957** | **0.557** | 0.574 | 0.929 | 0.637 | **0.546** |
| REFERENCE-FREE METRIC | | | | | | | | | | |
| GEMBA-MQM[†] | 0.802 | **0.993** | 0.502 | 0.572 | **0.939** | 0.401 | 0.564 | **0.991** | 0.449 | 0.522 |
| XCOMET-QE-Ensemble[†] | **0.808** | 0.974 | **0.679** | 0.588 | 0.909 | 0.498 | 0.554 | 0.892 | 0.647 | 0.533 |
| MetricX-23-QE-XXL | 0.797 | 0.934 | 0.547 | 0.607 | 0.813 | 0.459 | 0.575 | 0.877 | 0.652 | 0.531 |
| CometKiwi-QE-XL | 0.786 | 0.976 | 0.447 | 0.571 | 0.900 | 0.384 | 0.533 | 0.974 | 0.430 | 0.522 |
| ENSEMBLE BASELINES | | | | | | | | | | |
| Uniform | 0.801 | 0.935 | 0.552 | 0.600 | 0.816 | 0.484 | 0.566 | 0.948 | 0.637 | **0.540** |
| Weighted Avg | 0.802 | 0.934 | 0.555 | 0.603 | 0.811 | 0.487 | 0.568 | 0.943 | 0.647 | **0.540** |
| METAMETRICS-MT-QE | | | | | | | | | | |
| GP | 0.801 | 0.934 | 0.556 | **0.609** | 0.815 | 0.474 | **0.578** | 0.900 | **0.660** | 0.537 |
| XGBoost | 0.805 | 0.967 | 0.583 | 0.604 | 0.881 | **0.509** | 0.568 | 0.869 | 0.642 | 0.526 |

work Reward Data Collection[3] for training our METAMETRICS, including HelpSteer2 (Wang et al., 2024c), OffsetBias (Park et al., 2024), WildGuard (Han et al., 2024), and Magpie (Xu et al., 2024). In addition, we add Preference Test Data[4] to increase the size of the human preference dataset. We

---

[3]Cleaned Skywork Reward Data Collection can be accessed at `https://huggingface.co/datasets/natolambert/skywork-preferences-80k-v0.1-cleaned`.

[4]Preference Test Data from AI2 can be accessed at `https://huggingface.co/datasets/allenai/preference-test-sets`.

refer to our metric as METAMETRICS-RM and report the accuracy of predicting that the chosen sample is ranked higher than the rejected sample (Table 4).

## 5 RESULTS

In this section, we present the results of our METAMETRICS across five diverse tasks, covering both NLP and VL downstream applications.

Table 3: Kendall correlation results of METAMETRICS-QA. More detailed results for all metrics and their variants can be found in Table 11 in the Appendix. **Bold** and underlined values indicate the best and second best performance, respectively.

| Setup | EVOUNA | | EQAE | | Combined |
|---|---|---|---|---|---|
| | NQ | TQ | NarrativeQA | SemEval | |
| ARMORM METRICS | | | | | |
| ultrafeedback-honesty | 0.202 | 0.281 | 0.297 | 0.335 | 0.296 |
| helpsteer-helpfulness | 0.204 | 0.293 | 0.332 | 0.427 | 0.302 |
| helpsteer-correctness | 0.206 | 0.300 | 0.336 | 0.403 | 0.305 |
| argilla-judge-lm | 0.243 | 0.301 | 0.253 | 0.337 | 0.332 |
| N-GRAM-BASED METRICS | | | | | |
| BLEU1 | 0.461 | 0.353 | 0.362 | 0.260 | 0.424 |
| ROUGE-L | 0.495 | 0.399 | 0.584 | **0.518** | 0.454 |
| METEOR | 0.517 | 0.435 | 0.570 | 0.461 | 0.494 |
| ENSEMBLE BASELINES | | | | | |
| Uniform | 0.425 | 0.405 | 0.564 | 0.487 | 0.449 |
| Weighted Avg. | 0.497 | 0.423 | 0.582 | 0.467 | 0.484 |
| METAMETRICS-QA | | | | | |
| GP (Linear) | 0.518 | 0.448 | 0.586 | 0.464 | 0.512 |
| XGBoost | **0.543** | 0.471 | 0.601 | 0.486 | **0.536** |
| XGBoost (Iterative Top 5) | 0.540 | **0.488** | **0.626** | 0.503 | 0.528 |

**Abstractive Text Summarization.** For this task, we refer to our metric as METAMETRICS-SUM. Table 1 shows the results of abstractive text summarization task. Overall, our proposed model outperforms all other baselines, including all ensemble models and overall best automatic metric ($\Delta$ 0.023). The METAMETRICS-SUM with GP also performs considerably well, also outperforms all other baselines. Additionally, we present the results of BenchmarkLLM in Appendix Table 10.

**Machine Translation.** In this task, we demonstrate that METAMETRICS-MT in the reference-based setting outperforms all baseline models, including ensemble methods (Table 2). Notably, the top competitor, XCOMET-Ensemble, achieves impressive results on the WMT23 evaluation leaderboard, particularly excelling in the two-segment Pearson correlation metric for `en-de` and `zh-en`. However, METAMETRICS consistently surpasses XCOMET-Ensemble across all other tasks, leading to a higher average correlation overall, achieving the new state-of-the-art performance on WMT23 evaluation. In the reference-free setting, METAMETRICS-MT-QE performs comparably to XCOMET-QE-Ensemble. While XCOMET-Ensemble shows slightly better average correlation, our proposed model demonstrates superior performance on the acc-t metric. This is particularly significant, as the acc-t metric is finely tuned to align with the Kendall correlation $\tau$ objective function, indicating a stronger correlation with human evaluations.

**Question Answering.** Table 3 presents the results of METAMETRICS-QA, which consistently outperforms existing metrics across all evaluation levels. While ROUGE performs slightly better than our method in the SemEval dataset, our proposed model demonstrates a significant advantage over others in different datasets, achieving improvements of up to 0.044 points. Notably, our iterative-based approach enhances the performance of the vanilla METAMETRICS-QA when using XGB, all while utilizing fewer metrics. Also, for the GP evaluation, choosing either all metrics or just the top five results in similar combined scores, showing that our metric selection strategy is robust.

**Image Captioning.** Figure 2 presents the results of METAMETRICS-CAP and cross-dataset tuning. METAMETRICS-CAP consistently performs better compared to any individual metric, even

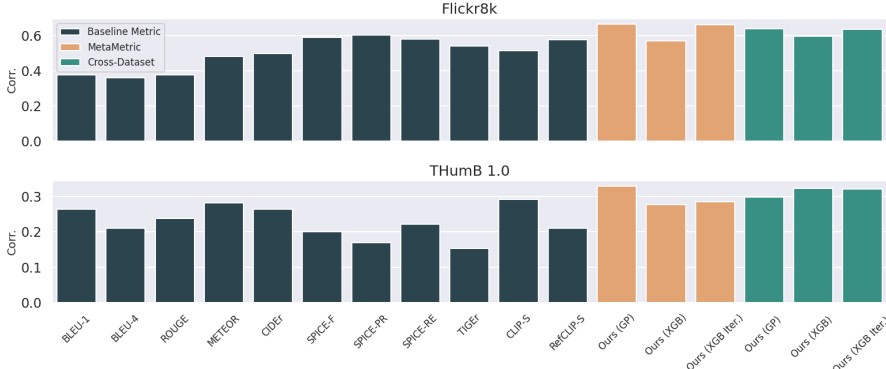

Figure 2: Kendall correlation results of METAMETRICS-CAP. Cross-Dataset means that METAMETRICS is tuned on THumB 1.0 and tested on Flickr8k, and vice versa. Our METAMETRICS-CAP outperforms other metrics across different datasets and shows robustness even when trained and tested on different datasets (Cross-Dataset setting). More detailed results for all metrics and their variants can be found in Table 13 in the Appendix.

when the tuning is performed on a dataset from a different distribution. Although tuning on the same distribution yields the best results, we observe that METAMETRICS-CAP still outperforms any individual metric, demonstrating its robustness across varying data distributions.

**Reward Model Scoring.** Table 4 highlights the performance of METAMETRICS-RM, which achieves state-of-the-art results for GP and competes effectively with the top seven metrics, including the advanced 70B Nemotron Reward model. These results demonstrate that our proposed metric is both efficient and effective, showcasing its robustness across diverse categories of tasks, datasets, and evaluation settings. Overall, METAMETRICS-RM represents a significant advancement in reward metric evaluation, offering high performance and broad applicability for researchers and practitioners alike.

Table 4: Accuracy results in percentages for Reward-Model-As-A-Metric. More detailed results for all metrics and their variants can be found in Table 14 in the Appendix. **Bold** and underlined values indicate the best and second best performance, respectively. Our METAMETRICS-RM achieves the best or comparable results in 2 out of the 4 categories in RewardBench.

| Metric | Score | Chat | Chat Hard | Safety | Reasoning |
|---|---|---|---|---|---|
| Llama-3.1-Nemotron-70B-Reward | **94.1** | 97.5 | 85.7 | **95.1** | 98.1 |
| Skywork-Reward-Gemma-2-27B | 93.8 | 95.8 | **91.4** | 91.9 | 96.1 |
| TextEval-Llama3.1-70B | 93.5 | 94.1 | 90.1 | 93.2 | 96.4 |
| Skywork-Critic-Llama-3.1-70B | 93.3 | 96.6 | 87.9 | 93.1 | 95.5 |
| SFR-LLaMa-3.1-70B-Judge-r | 92.7 | 96.9 | 84.8 | 91.6 | 97.6 |
| URM-LLaMa-3.1-8B | 92.9 | 95.5 | 88.2 | 91.1 | 97.0 |
| Skywork-Reward-Llama-3.1-8B | 92.5 | 95.8 | 87.3 | 90.8 | 96.2 |
| GRM-Llama3-8B | 91.5 | 95.5 | 86.2 | 90.8 | 93.6 |
| ArmoRM-Llama3-8B-v0.1 | 90.4 | 96.9 | 76.8 | 90.5 | 97.3 |
| URM-LLaMa-3-8B | 89.9 | 96.9 | 78.7 | 88.2 | 95.7 |
| ENSEMBLE BASELINES | | | | | |
| Uniform | 92.5 | 98.3 | 83.1 | 90.8 | 97.6 |
| Weighted Avg. | 92.5 | 98.3 | 83.1 | 90.7 | 97.6 |
| METAMETRICS-RM | | | | | |
| GP | 93.5 | **98.9** | 86.2 | 90.7 | **98.2** |
| XGBoost (Iterative Best) | 92.9 | 95.8 | 89.7 | 92.2 | 94.0 |

## 6 ANALYSIS AND DISCUSSION

In this section, we analyze the advantages of the methods in terms of interpretability, efficiency, and robustness. We discuss how METAMETRICS enhances the capability of model evaluation.

## 6.1 INTERPRETABILITY

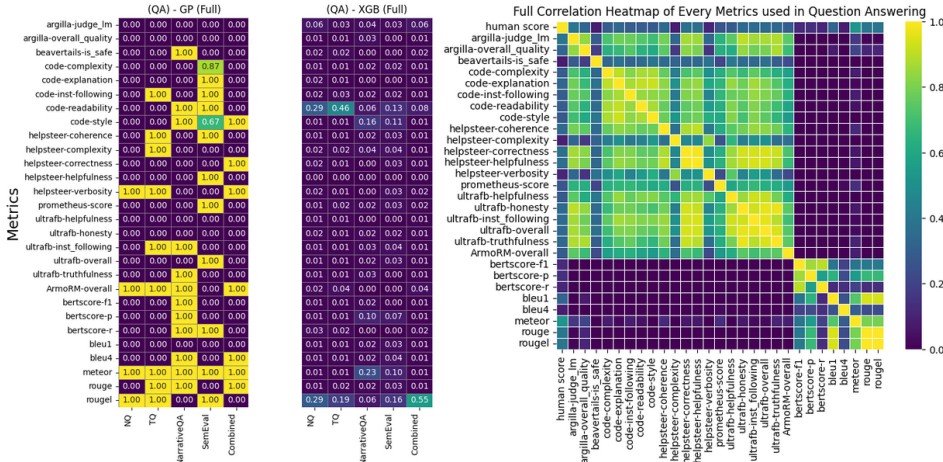

Figure 3: GP weights **(Left)** and XGBoost Features Importance **(Center)** with Intra-Metric Correlation **(Right)** for METAMETRICS-QA.

Interpretability is a key aspect in METAMETRICS, as the optimization process inherently reveals the impact of each metric on the final score. The chosen methods, such as feature importance in Boosting and weights analysis in Bayesian Optimization, provide a straightforward way to quantify each metric's contribution to human preference alignment. As shown in Figure 3, this interpretability is highlighted in the QA task, showcasing the importance of each metric in human preference alignment. For example, since ROUGE and ROUGE-L are two highly correlated metrics, Bayesian Optimization often drops one of them during the tuning process.

## 6.2 EFFECTIVENESS AND EFFICIENCY

**Improved Performance with Fewer Parameters.** Table 4 highlights the compute efficiency and superior performance of METAMETRICS-RM. METAMETRICS-RM, utilizing GP. Despite this relatively small compute memory footprint, it outperforms models with 70B parameters. This demonstrates that METAMETRICS-RM offers state-of-the-art accuracy using significantly more cost-effective models, making it an ideal choice for resource-constrained applications while delivering even better performance.

**High Parallelization.** METAMETRICS is designed to be embarrassingly parallel, enabling each metric to be executed independently without requiring inter-model communication, which results in exceptional efficiency. Unlike large models with 27B parameters that often face bottlenecks, our approach supports the development of highly effective models without such time constraints. Furthermore, METAMETRICS excels at selecting a sparse yet impactful set of metrics from a vast pool of potential evaluation metrics. As shown in Figure 3, both GP and XGBoost can identify a small subset of key metrics that strongly correlate with human preferences, thereby reducing the need to evaluate a wide range of metrics. This selective approach identifies essential metrics for specific tasks while minimizing computational overhead without compromising performance. In essence, METAMETRICS is ideally suited for low-resource settings, providing superior alignment with human preferences across multiple tasks.

## 6.3 ROBUSTNESS

We present our robustness evaluation in both cross-lingual and cross-dataset scenarios. Comprehensive experimental results are detailed in Subsection F.7. For the cross-lingual evaluation, we assess our method on WMT23 and WMT24 using unseen language pairs. Specifically, we evaluate METAMETRICS on he-en (Hebrew-English), en-es (English-Spanish), and ja-zh (Japanese-Chinese). Overall, our approach outperforms all existing baseline metrics. We also evaluate the robustness of METAMETRICS through cross-dataset experiments in image captioning, tuning on one dataset and testing on another. Despite variations in domain and content, our results consis-

tently demonstrate the effectiveness of METAMETRICS. Figure 2 shows that METAMETRICS-CAP tuned on THumB 1.0 and tested on Flickr8k and vice-versa out performs all individual metrics. This robustness indicates that METAMETRICS is not overfitting to specific dataset characteristics, but rather learning to combine metrics in a way that aligns with general human judgments of caption quality. This generalization capability is crucial for real-world applications, where the evaluation metric may need to perform well on diverse and potentially out-of-domain data.

# 7 RELATED WORK

Performance evaluation metrics for natural language generation tasks can be categorized into several categories based on how the metric compares generated texts with reference texts.

**Surface-level Metrics.**    These metrics compare the generated text to reference text at the word level, focusing on n-gram overlap or direct lexical matching. Common metrics include BLEU that measures precision of n-grams between the generated- and reference-texts (Papineni et al., 2002), ROUGE that measures recall (Lin, 2004), METEOR (Banerjee & Lavie, 2005) that goes beyond exact word overlap and considers stemming, synonyms, and paraphrasing, and chrF (Popović, 2015) that calculates similarity based on character n-grams rather than word n-grams using F-score.

**Embedding-based Metrics.**    These metrics rely on word or sentence embeddings to measure semantic similarity between generated and reference text, capturing deeper meaning rather than surface overlap. Examples include MoverScore (Zhao et al., 2019), which considers semantic similarity at the word level and accounts for word movement (alignment) between texts, and BERTScore (Zhang et al., 2019), a neural-based metric that compares semantic similarity and understands contextual relationships using BERT embeddings. COMET (Rei et al., 2022) and BLEURT (Sellam et al., 2020) are neural-based metrics that use contextualized embeddings from models like BERT as inputs to train models to generate prediction estimates of human judgments. For vision-language tasks, ClipScore (Hessel et al., 2021) uses embeddings to compare vision and language modalities.

**Ensemble-based Metrics.**    Ensemble-based metrics combine multiple individual metrics to enhance evaluation robustness through averaging (Adiwardana et al., 2020) or heuristics (Phy et al., 2020). The Absolute-Rating Multi-Objective Reward Model (ArmoRM) (Wang et al., 2024b) is a SotA reward model designed for RLHF (Ouyang et al., 2022).

**LLM-based Metrics.**    LLM-based metrics leverage generative language models to assess the quality of an input by generating scores. Examples include BARTScore (Yuan et al., 2021), G-Eval (Liu et al., 2023) utilizing the GPT-4 model (Achiam et al., 2023), Unieval (Zhong et al., 2022) utilizing the T5 model (Raffel et al., 2020), MetricX (Juraska et al., 2023; 2024), COMET, CometKiwi (Rei et al., 2022), XCOMET (Guerreiro et al., 2023), and GEMBA-MQM (Kocmi & Federmann, 2023).

# 8 CONCLUSION

Understanding the quality of a performance evaluation metric is crucial for ensuring alignment with human preferences. However, it remains unclear how well each metric captures the diverse aspects of these preferences, as metrics often excel in one particular area but not across all dimensions. To address this, it is essential to systematically calibrate metrics to specific aspects of human preference, catering to the unique characteristics of each aspect. We introduce METAMETRICS, a calibrated meta-metric designed to evaluate generation tasks across different modalities in a supervised manner. METAMETRICS optimizes the combination of existing metrics to enhance their alignment with human preferences. Our method demonstrates flexibility and effectiveness in both language and vision downstream tasks, showing significant benefits across various multilingual and multi-domain scenarios. Our proposed metric aligns closely with human preferences and it is also highly extendable and easily integrable into any application. This makes it a powerful tool for improving the evaluation of generation tasks, ensuring that metrics are more representative of human judgment across diverse contexts.

## ACKNOWLEDGMENTS

We would like to express our sincere gratitude to Mingyang Zhou for his insightful suggestions and discussions on this project.

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

## A  LIMITATIONS AND FUTURE WORK

In our experiment, we utilize metrics that are commonly employed to evaluate each downstream task. However, due to limitations in computational resources, we exclude LLMs exceeding 10.7B parameters to ensure that our metrics can be executed on commercial GPU resources with a maximum memory capacity of 48GB. Additionally, we restrict our exploration of metrics to avoid exhaustively incorporating LLM-based metrics, given our capacity and resource constraints. Furthermore, we leave the exploration of RLHF with PPO to future studies.

Looking ahead, there are significant opportunities for further exploration. We can explore into metrics across various promising research avenues, such as additional languages, expanded vision and speech modalities (Ho et al., 2020; Blattmann et al., 2023; Zheng et al., 2024), and larger multilingual generation tasks. In particular, we identify METAMETRICS as a valuable tool for multilingual (Winata et al., 2019a; Joshi et al., 2020; Anugraha et al., 2024), multi-cultural (Adilazuarda et al., 2024; Winata et al., 2024a; Yue et al., 2024), and code-switching (Winata et al., 2019b; 2023; Kuwanto et al., 2024) applications.

## B  METRICS

In this section, we list all metrics we use in our experiments.

### B.1  ABSTRACTIVE TEXT SUMMARIZATION

We use the following metrics:

**n-Gram-based Metrics:** BLEU-4 (Papineni et al., 2002), chrF (Popović, 2015), METEOR (Banerjee & Lavie, 2005), ROUGE-1, ROUGE-3 , ROUGE-4, ROUGE-L (Lin, 2004), ROUGE-WE1 (Ng & Abrecht, 2015).

**Embedding-based Metrics:** BARTScore (Yuan et al., 2021), BLEURT (Sellam et al., 2020), BERTScore (Zhang et al., 2019).

**LLM-based Metrics:** SummaQA (Scialom et al., 2019), UniEval (Zhong et al., 2022), G-Eval (Liu et al., 2023).

## B.2 MACHINE TRANSLATION

We use the following metrics:

**n-Gram-based Metrics:** chrF (Popović, 2015), BLEU (Papineni et al., 2002);

**Embedding-based Metrics:** BERTScore (Zhang et al., 2019), Yisi-1 (Lo, 2020);

**LLM-based Metrics:** MetricX-23 (L, XL, XXL) (Juraska et al., 2023), COMET, CometKiwi (Rei et al., 2022), XCOMET (Guerreiro et al., 2023), GEMBA-MQM (Kocmi & Federmann, 2023).

For machine translation, we optimize the Kendall correlation during our tuning.

## B.3 QUESTION ANSWERING

We use the following metrics:

**n-Gram-based Metrics:** BLEU-1, BLEU-4 BLEU (Papineni et al., 2002), ROUGE, ROUGE-L (Lin, 2004), METEOR (Banerjee & Lavie, 2005);

**Embedding-based Metrics:** BERTScore (Zhang et al., 2019);

**LLM-based Metrics:** ArmoRM-Llama3-8B-v0.1 (ArmoRM) (Wang et al., 2024b).

## B.4 IMAGE CAPTIONING

We use the following metrics:

**n-Gram based metrics** BLEU (Papineni et al., 2002), ROUGE (Lin, 2004), METEOR (Banerjee & Lavie, 2005), SPICE (Anderson et al., 2016), CIDEr (Vedantam et al., 2015);

**Embedding-based Metrics** TIGEr (Jiang et al., 2019) and CLIPScore (Both reference-based and reference-free) (Hessel et al., 2021).

## B.5 REWARD MODEL SCORING

The reward model used in the experiment is listed in Table 5.

Table 5: Reward Models.

| Reward Model / Metric | Hugging Face Link |
| --- | --- |
| nvidia/Llama-3.1-Nemotron-70B-Reward | https://huggingface.co/nvidia/Llama-3.1-Nemotron-70B-Reward |
| SF-Foundation/TextEval-Llama3.1-70B | https://huggingface.co/SF-Foundation/TextEval-Llama3.1-70B |
| Skywork/Skywork-Critic-Llama-3.1-70B | https://huggingface.co/Skywork/Skywork-Critic-Llama-3.1-70B |
| Salesforce/SFR-LLaMa-3.1-70B-Judge-r | https://huggingface.co/Salesforce/SFR-LLaMa-3.1-70B-Judge-r |
| internlm/internlm2-1_8b-reward (Cai et al., 2024) | https://huggingface.co/internlm/internlm2-1_8b-reward |
| internlm/internlm2-7b-reward (Cai et al., 2024) | https://huggingface.co/internlm/internlm2-7b-reward |
| LxzGordon/URM-LLaMa-3-8B | https://huggingface.co/LxzGordon/URM-LLaMa-3-8B |
| LxzGordon/URM-LLaMa-3.1-8B | https://huggingface.co/LxzGordon/URM-LLaMa-3.1-8B |
| Ray2333/GRM-Gemma-2B-rewardmodel-ft | https://huggingface.co/Ray2333/GRM-Gemma-2B-rewardmodel-ft |
| Ray2333/GRM-Llama3-8B-rewardmodel-ft | https://huggingface.co/Ray2333/GRM-Llama3-8B-rewardmodel-ft |
| sfairXC/FsfairX-LLaMA3-RM-v0.1 | https://huggingface.co/sfairXC/FsfairX-LLaMA3-RM-v0.1 |
| Skywork/Skywork-Reward-Llama-3.1-8B | https://huggingface.co/Skywork/Skywork-Reward-Llama-3.1-8B |
| weqweasdas/RM-Mistral-7B | https://huggingface.co/weqweasdas/RM-Mistral-7B |

## C METRIC DESIGN

### C.1 FRAMEWORK

Figure 4 illustrates the framework for both reference-free and reference-based settings.

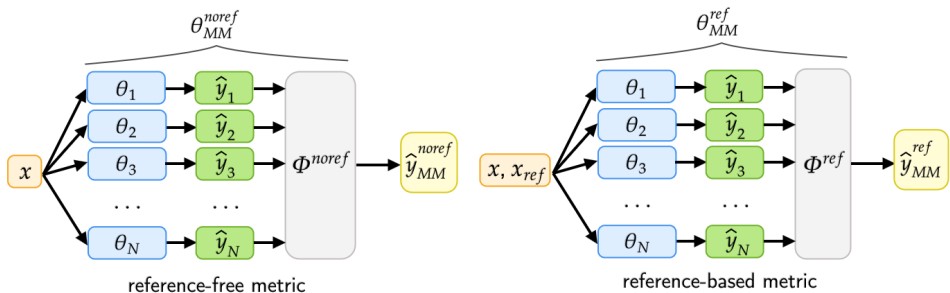

Figure 4: METAMETRICS Framework for reference-free ($\theta_{\text{MM}}^{\text{noref}}$, left) and reference-based setting ($\theta_{\text{MM}}^{\text{ref}}$, right). METAMETRICS integrates multiple metrics $\theta_i$ and their scores $\hat{y}_i$, learns a function $\Phi$ to combine them into a score $\hat{y}_{\text{MM}}$ that aligns best with the human judgment score.

## C.2 ALTERNATIVE WEIGHTING SCORE

**Multiplicative GP** Recall that in the linear-weighting scheme for GP, given a set of $N$ evaluation metrics $\hat{\mathbf{y}} = [\hat{y}_1, \hat{y}_2, \ldots, \hat{y}_N]$ with their associated weights of $\{w_1, w_2, \ldots, w_N\}$, GP linear is given by $\hat{y}_{\text{MM-LINEAR}}(\mathbf{w}) = \mathbf{w}^\top \hat{\mathbf{y}}$.

To explore more weighting schemes, we introduce "Multiplicative" GP, which accounts for interactions between metrics through pairwise products. For $N$ metrics, there are $c = \binom{N}{2} = \frac{N \cdot (N-1)}{2}$ unique pairwise combinations. Let $\mathbf{w}_{\text{pair}}$ be the weight vector for these combinations, defined as $\mathbf{w}_{\text{pair}} = [w_1, w_2, \ldots, w_c]$ and let $\hat{\mathbf{y}}_{\text{pair}}$ be the vector of pairwise products, where $\hat{\mathbf{y}}_{\text{pair}} = [\hat{y}_i \hat{y}_j \mid 1 \leq i < j \leq N]$. The Multiplicative GP output is then computed as $\hat{y}_{\text{MM-multi}}(\mathbf{w}_{\text{pair}}) = \mathbf{w}_{\text{pair}}^\top \hat{\mathbf{y}}_{\text{pair}}$.

## D DATASETS

In this section, we present the data statistics and outline the preprocessing procedures used in our work.

### D.1 DATA STATISTICS

Table 6 presents the dataset split sizes for tuning and testing.

### D.2 DATA PREPROCESSING

**Pre-processing.** We apply pre-processing before we use the scores in the training. The pre-processing can be defined as follows:

1. **Clipping:** Let the valid range for $\hat{y}_i$ be defined by $[\hat{y}_i^{\min}, \hat{y}_i^{\max}]$. The clipped metric $\hat{y}_i'$ can be defined as:

$$\hat{y}_i' = \begin{cases} \hat{y}_i^{\min} & \text{if } \hat{y}_i < \hat{y}_i^{\min}, \\ \hat{y}_i & \text{if } \hat{y}_i^{\min} \leq \hat{y}_i \leq \hat{y}_i^{\max}, \\ \hat{y}_i^{\max} & \text{if } \hat{y}_i > \hat{y}_i^{\max}. \end{cases} \tag{2}$$

2. **Normalization:** After clipping, the score is normalized to a common scale of $[0, 1]$:

$$\hat{y}_i = \frac{\hat{y}_i' - \hat{y}_i^{\min}}{\hat{y}_i^{\max} - \hat{y}_i^{\min}}. \tag{3}$$

3. **Inversion (if applicable):** If the metric is such that higher scores indicate worse performance, we invert the normalized score:

$$\hat{y}_i = 1 - \hat{y}_i. \tag{4}$$

Table 6: Dataset statistics used in the experiments.

| Dataset Name | Type | Tuning Size | Testing Size |
|---|---|---|---|
| **Machine Translation** | | | |
| WMT23 (MQM) (Freitag et al., 2023) | Translation | - | 136,260 |
| WMT20-22 (MQM) (Freitag et al., 2022) | Translation | 150,343 | - |
| **Text Summarization** | | | |
| SummEval (Fabbri et al., 2021) | Summarization | 510 | 1,190 |
| BenchmarkLLM (Zhang et al., 2024) | Summarization | 1,035 | 2,415 |
| Combined | Summarization | 1,545 | 3,605 |
| **Question Answering** | | | |
| NQ (Kwiatkowski et al., 2019) | Open-QA | 4,528 | 10,567 |
| TQ (Joshi et al., 2017) | Open-QA | 2,906 | 6,783 |
| NarrativeQA (Kočiskỳ et al., 2018) | RCQA | 150 | 350 |
| SemEval (Ostermann et al., 2018) | Reasoning QA | 90 | 210 |
| Combined | Multidomain-QA | 7,675 | 17,909 |
| **Image Captioning** | | | |
| Flickr8k-Expert (Hodosh et al., 2013) | Captioning | 1,746 | 4,076 |
| THumB 1.0 (Kasai et al., 2022) | Captioning | 600 | 1,400 |
| **Reward Model Scoring** | | | |
| Skywork-Reward-Preference-80K-v0.1 | Math, Code, Chat | 81,973 | - |
| allenai/preference-test-sets | Math, Code, Chat | 43,175 | - |
| RewardBench (Lambert et al., 2024) | Chat, Safety, Reasoning | - | 8,108 |

# E OPTIMIZATION

In this section, we explore the optimization details of METAMETRICS.

## E.1 METHOD DETAILS

### E.1.1 MATERN KERNEL

We train BO using GP with Matern kernel, a generalization of the RBF, distinguished by an additional parameter that controls the smoothness of the resulting function (Williams & Rasmussen, 2006). Conversely, as the parameter approaches infinity, the Matern kernel converges to the RBF kernel. The kernel is described as below:

$$k(\mathbf{w}, \mathbf{w}') = \frac{1}{\Gamma(\nu)2^{\nu-1}} \left( \frac{\sqrt{2\nu}}{l} d(\mathbf{w}, \mathbf{w}') \right)^\nu K_\nu \left( \frac{\sqrt{2\nu}}{l} d(\mathbf{w}, \mathbf{w}') \right), \tag{5}$$

where $d(\cdot, \cdot)$ is the Euclidean distance, $K_\nu(\cdot)$ is a modified Bessel function and $\Gamma(\cdot)$ is the gamma function.

### E.1.2 ITERATIVE-BASED PRUNING

Below is the implementation of Iterative-based Pruning performed during training.

## E.2 HYPER-PARAMETERS

### E.2.1 BAYESIAN OPTIMIZATION

Table 7 describes the hyper-parameter settings that we use for our experiments. For the Bayesian optimization, we run GP with a Matern kernel Williams & Rasmussen (2006), a generalization of the RBF kernel, using $\nu = 2.5$.

---

**Algorithm 1** Iterative-based Pruning with XGBoost

---

1: **procedure** ITERATIVEXGBOOST($\mathbf{X}, \mathbf{y}, k$)
2:      $\mathcal{F} \leftarrow \{f_1, f_2, \ldots, f_p\}$            $\triangleright$ Initial set of features
3:      $\mathcal{P} \leftarrow []$            $\triangleright$ Performance history
4:      $\mathcal{F}_{\text{least}} \leftarrow []$            $\triangleright$ Feature pruning history
5:      **for** $i \leftarrow 1$ to $k$ **do**
6:          Train $\Phi_{\text{XGB}}^{(i)}$ on $\mathbf{X}_{\mathcal{F}}$ with CV
7:          $\mathcal{I}_i \leftarrow \text{Importance}(\Phi_{\text{XGB}})$            $\triangleright$ Compute feature importance from $\Phi_{\text{XGB}}$
8:          $\mathcal{P}[i] \leftarrow \rho(\Phi_{\text{XGB}}(\mathbf{X}_{\mathcal{F}}), \mathbf{y})$            $\triangleright$ Store performance score
9:          $f_{\text{least}}[i] \leftarrow \text{argmin}(\mathcal{I}_i)$            $\triangleright$ Identify least important feature
10:         $\mathcal{F}_{\text{least}} \leftarrow f_{\text{least}}$
11:         $\mathcal{F} \leftarrow \mathcal{F} \setminus \{f_{\text{least}}\}$            $\triangleright$ Remove least important feature
12:      **end for**
13:      $i^* \leftarrow \text{argmax}(\mathcal{P})$            $\triangleright$ Find the best index iteration
14:      $\mathcal{F}_{\text{best}} \leftarrow \mathcal{F} \cup \mathcal{F}_{\text{least}}[i^* :]$            $\triangleright$ Best features from highest performance
15:      Train final $\hat{f}_{\text{XGB}}$ on $\mathbf{X}_{\mathcal{F}_{\text{best}}}$
16:      **return** $\hat{f}_{\text{xgb}}$
17: **end procedure**

---

Table 7: Hyper-parameters used for Gaussian Process on each task.

| Metric | Hyper-parameter | Value |
|---|---|---|
| METAMETRICS-SUM | init_points | 5 |
| | n_iter | 100 |
| METAMETRICS-MT | init_points | 5 |
| | n_iter | 100 |
| METAMETRICS-QA | init_points | 5 |
| | n_iter | 40 |
| METAMETRICS-CAP | init_points | 5 |
| | n_iter | 100 |
| METAMETRICS-RM | init_points | 5 |
| | n_iter | 100 |

### E.2.2 BOOSTING METHOD

For XGBoost training, we use a different objective depending on the task. We perform parameter searching with hyper-parameter values in Table 8.

## F DETAILED RESULTS

In this section, we provide more detailed results that could not be included in the main paper due to space constraints.

### F.1 ABSTRACTIVE TEXT SUMMARIZATION

Table 9 provides a detailed breakdown of the correlations between various metrics and human preference measurements for abstractive text summarization on SummEval LLM. Table 10 provides a detailed breakdown of the correlations between various metrics and human preference measurements for abstractive text summarization on Benchmark LLM.

Table 8: Initial hyper-parameter values used for parameter searching during XGBoost.

| Metric | Hyper-parameter | Value |
|---|---|---|
| METAMETRICS-SUM | n_estimators_low | 100 |
| | n_estimators_high | 1000 |
| | n_estimators_step | 100 |
| | n_estimators_prior | Uniform |
| | objective | reg:squarederror |
| METAMETRICS-MT | n_estimators_low | 100 |
| | n_estimators_high | 1000 |
| | n_estimators_step | 100 |
| | n_estimators_prior | Uniform |
| | objective | reg:absoluteerror |
| METAMETRICS-QA | n_estimators_low | 100 |
| | n_estimators_high | 400 |
| | n_estimators_step | 25 |
| | n_estimators_prior | Uniform |
| | objective | reg:squaredlogerror |
| METAMETRICS-CAP | n_estimators_low | 100 |
| | n_estimators_high | 1000 |
| | n_estimators_step | 100 |
| | n_estimators_prior | Uniform |
| | objective | reg:squaredlogerror |
| METAMETRICS-RM | n_estimators_low | 100 |
| | n_estimators_high | 1000 |
| | n_estimators_step | 100 |
| | n_estimators_prior | Uniform |
| | objective | rank:pairwise |

### F.2 MACHINE TRANSLATION

Table 12 describes the detailed breakdown of the correlations between metrics with human annotators in WMT23.

### F.3 QUESTION ANSWERING

Table 11 provides a detailed breakdown of the correlations between various metrics and human ratings for the Question Answering task.

### F.4 IMAGE CAPTIONING

Table 13 provides a detailed breakdown of the correlations between various metrics and human annotations for Image Captioning.

### F.5 REWARD MODEL SCORING

Table 14 describes the detailed breakdown of accuracy between metrics with human preference on reward model scoring.

### F.6 ABLATION STUDY

Table 15 presents the results of an ablation study on the application of GP using three different weighting methods: linear, multiplicative, and combined (linear and multiplicative) on Summarization, Machine Translation, and Reward Model tasks as described in C. The results indicate that the linear weighting method generally outperforms the other approaches on average. This is likely due

Table 9: Kendall and spearman correlation detailed results with human ratings for summarization task on SummEval. **Coh.**, **Cons.**, **Fluency**, and **Rel.** corresponds to coherence, consistency, and fluency, relevance, respectively. **Bold** and underlined values indicate the best and second best performance, respectively.

| Metric | | | Kendall | | | | | Spearman | | |
|---|---|---|---|---|---|---|---|---|---|---|
| | Coh. | Cons. | Fluency | Rel. | Avg. | Coh. | Cons. | Fluency | Rel. | Avg. |
| BLEU | 0.110 | 0.126 | 0.113 | 0.170 | 0.130 | 0.157 | 0.160 | 0.145 | 0.239 | 0.175 |
| chrF | 0.143 | 0.094 | 0.071 | 0.198 | 0.127 | 0.205 | 0.119 | 0.091 | 0.278 | 0.173 |
| METEOR | 0.077 | 0.102 | 0.072 | 0.162 | 0.103 | 0.108 | 0.130 | 0.093 | 0.229 | 0.140 |
| ROUGE-1 | 0.112 | 0.103 | 0.061 | 0.189 | 0.116 | 0.159 | 0.130 | 0.079 | 0.264 | 0.158 |
| ROUGE-3 | 0.099 | 0.115 | 0.048 | 0.154 | 0.104 | 0.142 | 0.147 | 0.063 | 0.219 | 0.143 |
| ROUGE-4 | 0.096 | 0.122 | 0.067 | 0.143 | 0.107 | 0.137 | 0.155 | 0.087 | 0.203 | 0.146 |
| ROUGE-l | 0.110 | 0.087 | 0.069 | 0.157 | 0.106 | 0.156 | 0.111 | 0.089 | 0.219 | 0.144 |
| ROUGE-WE1 | 0.115 | 0.088 | 0.081 | 0.169 | 0.113 | 0.164 | 0.112 | 0.105 | 0.237 | 0.115 |
| BLEURT (Max) | 0.185 | 0.070 | 0.114 | 0.189 | 0.140 | 0.262 | 0.088 | 0.062 | 0.194 | 0.152 |
| BLEURT (Mean) | 0.099 | 0.011 | 0.029 | 0.130 | 0.067 | 0.123 | 0.094 | 0.052 | 0.202 | 0.118 |
| BERTScore (f1) | 0.105 | 0.100 | 0.120 | 0.181 | 0.127 | 0.150 | 0.128 | 0.155 | 0.256 | 0.172 |
| BERTScore (Recall) | 0.104 | 0.111 | 0.093 | 0.176 | 0.121 | 0.148 | 0.141 | 0.120 | 0.250 | 0.165 |
| SummaQA (f1) | 0.051 | 0.120 | 0.106 | 0.114 | 0.098 | 0.073 | 0.152 | 0.136 | 0.165 | 0.132 |
| SummaQA (Prob) | 0.070 | 0.100 | 0.070 | 0.138 | 0.095 | 0.098 | 0.128 | 0.092 | 0.196 | 0.129 |
| LLM-BASED METRICS | | | | | | | | | | |
| BARTScore (Mean) | 0.086 | 0.074 | 0.040 | 0.143 | 0.086 | 0.123 | 0.094 | 0.052 | 0.202 | 0.118 |
| UniEval | 0.413 | 0.353 | 0.359 | 0.324 | 0.362 | 0.577 | 0.439 | 0.458 | 0.446 | 0.480 |
| G-Eval (GPT4) | 0.429 | 0.413 | 0.409 | 0.437 | 0.422 | 0.565 | 0.510 | 0.470 | 0.581 | 0.531 |
| ENSEMBLE BASELINES | | | | | | | | | | |
| Uniform | 0.141 | 0.133 | 0.117 | 0.213 | 0.151 | 0.201 | 0.170 | 0.150 | 0.298 | 0.205 |
| Weighted Avg | 0.150 | 0.141 | 0.122 | 0.220 | 0.159 | 0.215 | 0.179 | 0.158 | 0.309 | 0.215 |
| METAMETRICS-SUM | | | | | | | | | | |
| GP | 0.172 | 0.140 | 0.130 | 0.252 | 0.174 | 0.244 | 0.179 | 0.167 | 0.354 | 0.236 |
| XGBoost | 0.192 | 0.186 | 0.186 | 0.276 | 0.210 | 0.274 | 0.236 | 0.239 | 0.386 | 0.284 |
| W/ LLM-BASED METRICS | | | | | | | | | | |
| GP | 0.454 | 0.419 | 0.409 | **0.449** | 0.433 | 0.609 | 0.519 | 0.470 | **0.601** | 0.550 |
| GP (Top 2) | 0.461 | 0.428 | 0.409 | **0.449** | 0.437 | 0.628 | **0.528** | 0.470 | **0.601** | 0.557 |
| XGBoost | **0.476** | 0.367 | 0.404 | 0.447 | 0.424 | **0.642** | 0.460 | 0.512 | 0.600 | 0.553 |
| XGBoost (Top 2) | **0.476** | **0.436** | **0.430** | 0.445 | **0.447** | 0.636 | 0.508 | 0.511 | 0.594 | **0.562** |

to GP's preference for operating on lower-dimensional spaces (Li et al., 2016; Frazier, 2018; Nayebi et al., 2019; Malu et al., 2021; Binois & Wycoff, 2022). These findings reiterate the benefits of using fewer metrics for evaluation, which simplifies the optimization process.

Table 16 compares Kendall and Spearman correlations with human ratings for the summarization task using XGBoost. The results show minimal performance differences between the two objective functions, which is expected given the similarity of evaluation measurement between Spearman and Kendall correlations.

## F.7  ROBUSTNESS

We demonstrate the robustness of METAMETRICS in two scenarios: cross-lingual and cross-dataset. The cross-lingual setting assesses our method's capability with unseen languages, while the cross-dataset setting evaluates its performance in the face of distribution shifts or when the dataset originates from a different domain.

### F.7.1  CROSS-LINGUAL SETTING

We evaluate our method on WMT23 and WMT24 for unseen language pairs. To calibrate METAMETRICS, we employ three specific language pairs: zh-en (Chinese-English), en-de (English-German), and en-ru (English-Russian). For WMT23, we test our calibrated metric on unseen language pairs, such as he-en (Hebrew-English), with results presented in Table 17. For WMT24, we extend the evaluation to include en-es (English-Spanish) and ja-zh (Japanese-Chinese), with results detailed in Table 18.

Table 10: Kendall correlation results with human ratings for summarization task on Benchmark LLM. **Faith.**, **Coh.**, and **Cons.** correspond to faithfulness, coherence, and consistency, respectively. **Bold** and underlined values indicate the best and second best performance, respectively.

| Metric | Benchmark LLM | | |
| --- | --- | --- | --- |
| | Faith. | Coh. | Cons. |
| BLEU | 0.162 | 0.246 | 0.252 |
| chrF | 0.272 | 0.163 | 0.342 |
| METEOR | 0.251 | 0.182 | 0.331 |
| ROUGE-1 | 0.195 | 0.252 | 0.290 |
| ROUGE-3 | 0.138 | 0.142 | 0.225 |
| ROUGE-4 | 0.138 | 0.085 | 0.200 |
| ROUGE-l | 0.137 | 0.270 | 0.239 |
| ROUGE-WE1 | 0.223 | 0.218 | 0.292 |
| BARTScore (Mean) | 0.080 | 0.216 | 0.166 |
| BLEURT (Max) | 0.046 | 0.264 | 0.143 |
| BLEURT (Mean) | 0.046 | 0.264 | 0.143 |
| BERTScore (f1) | 0.171 | 0.280 | 0.283 |
| BERTScore (Recall) | 0.228 | 0.223 | 0.333 |
| SummaQA (f1) | 0.318 | 0.038 | 0.293 |
| SummaQA (Prob) | 0.330 | -0.003 | 0.303 |
| ENSEMBLE BASELINES | | | |
| Uniform | 0.226 | 0.234 | 0.328 |
| Weighted Avg | 0.256 | 0.264 | 0.337 |
| METAMETRICS-SUM | | | |
| GP | 0.356 | 0.308 | 0.383 |
| XGBoost | **0.406** | **0.411** | **0.397** |
| XGBoost (Iterative Top 5) | 0.377 | 0.378 | 0.383 |

### F.7.2 CROSS-DATASET SETTING

In addition to the cross-lingual setting, we also explore a cross-dataset scenario specifically for image captioning, as shown in Table 19. Here, we present the results when METAMETRICS-CAP is tuned on THuMB 1.0 and tested on Flickr8k, and vice versa. Our findings reveal that METAMETRICS-CAP, whether using GP or XGBoost, outperforms individual metrics, demonstrating its robustness in the presence of data shifts.

### F.8 TIME BENCHMARK

We benchmark the time elapsed to run training and performing inference. All experiments are run on the same machine with AMD EPYC 9354 32-Core Processor and NVIDIA RTX 6000 Ada GPU with 48GB memory. We show the detailed runtime for train and test split, and calibration on SummEval dataset in Table 20. Specifically, for G-Eval (GPT4), it costs around US$82 to run SummEval benchmark with a max token of 5, temperature of 2, top p of 1, a frequency penalty of 0, a presence penalty of 0, and completion choices of 20.

To compare the proper time evaluation on test split with METAMETRICS-SUM, it is important to account for all the components of METAMETRICS-SUM's evaluation process. Specifically, for individual metrics, we can consider the inference time on the test split. On the other hand, for METAMETRICS-SUM, we need to consider the total time, consisting of inference time on both the train and test splits, and then the calibration time using the train set. For example, G-Eval (GPT4) takes 7,173.50 seconds to evaluate on the test split, while XGBoost (all metrics including LLM-based metrics) takes $3,976.47 + 9,292.83 + 647.08 = 13,916.38$ seconds to evaluate on the test split.

Certain metric groups, such as BLEURT, SummaQA, and BERTScore, have multiple variations, all of which can be computed in a single evaluation pass. Therefore, the time measurement for METAMETRICS-SUM reflects the total duration of the pass, rather than the summation of time of all individual rows.

Table 11: Full performance of METAMETRICS-QA compared to the 8 best-performing standalone metrics and other baseline setups. **Bold** and underlined values indicate the best and second best performance, respectively. The **Combined** column reflects results when all datasets are merged.

| Setup | EVOUNA | | EQAE | | Combined |
|---|---|---|---|---|---|
| | NQ | TQ | NarrativeQA | SemEval | |
| ARMORM METRICS | | | | | |
| ultrafeedback-honesty | 0.202 | 0.281 | 0.297 | 0.335 | 0.296 |
| helpsteer-helpfulness | 0.204 | 0.293 | 0.332 | 0.427 | 0.302 |
| helpsteer-correctness | 0.206 | 0.300 | 0.336 | 0.403 | 0.305 |
| argilla-judge-lm | 0.243 | 0.301 | 0.253 | 0.337 | 0.332 |
| N-GRAM-BASED METRICS | | | | | |
| BLEU1 | 0.461 | 0.353 | 0.362 | 0.260 | 0.424 |
| ROUGE | 0.494 | 0.399 | 0.586 | **0.521** | 0.454 |
| ROUGE-L | 0.495 | 0.399 | 0.584 | 0.518 | 0.454 |
| METEOR | 0.517 | 0.435 | 0.570 | 0.461 | 0.494 |
| ENSEMBLE BASELINES | | | | | |
| Uniform | 0.425 | 0.405 | 0.564 | 0.487 | 0.449 |
| Weighted Avg. | 0.497 | 0.423 | 0.582 | 0.467 | 0.484 |
| METAMETRICS-QA | | | | | |
| GP (All) | 0.518 | 0.448 | 0.586 | 0.464 | 0.512 |
| GP (Top 5) | 0.506 | 0.450 | 0.570 | 0.452 | 0.512 |
| XGBoost (All) | 0.543 | 0.471 | 0.601 | 0.486 | 0.536 |
| XGBoost (Top 5) | 0.535 | **0.491** | 0.624 | 0.513 | 0.534 |
| XGBoost (Iterative Best) | **0.545** | 0.488 | **0.626** | 0.510 | **0.538** |
| XGBoost (Iterative Top 5) | 0.540 | 0.488 | **0.626** | 0.503 | 0.528 |

An important observation is that the bottleneck for METAMETRICS-SUM lies in the inference time for both the train and test splits, which dominates the total runtime compared to the calibration time. This is particularly evident when METAMETRICS-SUM includes metrics like G-Eval. The calibration time is relatively minor, with the longest calibration being for GP (all metrics with LLM-based metrics), which takes 647.08 seconds. However, if the metric outputs are already available, the inference step can be skipped.

Currently, the reported time for processing the train and test splits for METAMETRICS-SUM assumes a sequential evaluation of metrics. However, this sequential approach can be optimized by running metrics in parallel, as their evaluations are completely independent of each other. In other words, these computations are embarrassingly parallelizable. With this optimization, the total runtime for METAMETRICS-SUM would effectively be the time taken by the slowest metric, plus the calibration time. For example, for GP (all metrics except LLM-based metrics), the longest metric that takes to be evaluated is SummaQA, which takes 790.11 seconds. Therefore, the total running time when running METAMETRICS-NUM in a completely parallel manner will be $790.11 + 402.51 = 1,192.62$ seconds.

# G WEIGHTS AND FEATURE IMPORTANCE

## G.1 QUESTION ANSWERING

There are 8 setups for the question-answering task. The resulting weights or feature importance for each setup are reported in Figure 5.

**GP-ALL (NQ).** The selected metrics are `helpsteer-verbosity`, `ArmoRM-overall`, `meteor`, and `rougel`.

Table 12: Correlation with human ratings on WMT23 (MQM). [†]The results are collected from Freitag et al. (2023). **Bold** and underlined values indicate the best and second best performance, respectively.

| Metric | overall | en-de | | | he-en | | | zh-en | | |
|---|---|---|---|---|---|---|---|---|---|---|
| | sys/seg avg-corr | sys pearson | seg pearson | seg acc-t | sys pearson | seg pearson | seg acc-t | sys pearson | seg pearson | seg acc-t |
| REFERENCE-BASED METRIC | | | | | | | | | | |
| chrF[†] | 0.694 | 0.866 | 0.232 | 0.519 | 0.776 | 0.221 | 0.460 | 0.809 | 0.063 | 0.485 |
| BLEU[†] | 0.696 | 0.917 | 0.192 | 0.520 | 0.769 | 0.220 | 0.442 | 0.734 | 0.119 | 0.472 |
| BERTScore[†] | 0.742 | 0.891 | 0.325 | 0.528 | 0.895 | 0.335 | 0.515 | 0.810 | 0.236 | 0.499 |
| Yisi-1[†] | 0.754 | 0.925 | 0.366 | 0.542 | 0.917 | 0.395 | 0.529 | 0.823 | 0.290 | 0.504 |
| MetricX-23-XXL[†] | 0.808 | 0.977 | 0.585 | 0.603 | 0.910 | 0.548 | 0.577 | 0.873 | 0.625 | 0.531 |
| XCOMET-Ensemble[†] | 0.825 | 0.980 | **0.695** | 0.604 | 0.950 | 0.556 | 0.586 | 0.927 | **0.650** | 0.543 |
| COMET | 0.779 | 0.990 | 0.432 | 0.575 | 0.940 | 0.401 | 0.531 | 0.898 | 0.396 | 0.514 |
| XCOMET-XL | 0.817 | 0.970 | 0.670 | 0.596 | 0.949 | 0.530 | 0.576 | 0.928 | 0.607 | 0.530 |
| XCOMET-XXL | 0.817 | 0.983 | 0.660 | 0.602 | 0.952 | 0.465 | 0.560 | **0.965** | 0.597 | **0.546** |
| ENSEMBLE BASELINES | | | | | | | | | | |
| Uniform | 0.824 | 0.989 | 0.688 | 0.610 | 0.954 | 0.546 | 0.588 | 0.935 | 0.641 | 0.543 |
| Weighted Avg | **0.827** | 0.989 | 0.690 | 0.613 | 0.955 | 0.545 | 0.587 | 0.937 | 0.642 | 0.545 |
| METAMETRICS-MT | | | | | | | | | | |
| GP | 0.819 | 0.970 | 0.638 | 0.610 | 0.947 | 0.546 | **0.590** | 0.900 | 0.646 | 0.539 |
| XGBoost | 0.825 | **0.992** | 0.680 | **0.616** | **0.957** | **0.557** | 0.574 | 0.929 | 0.637 | **0.546** |
| REFERENCE-FREE METRIC | | | | | | | | | | |
| mbr-metricx-qe[†] | 0.788 | 0.976 | 0.571 | 0.584 | 0.915 | 0.411 | 0.553 | 0.936 | 0.489 | 0.537 |
| MetricX-23-QE[†] | 0.800 | 0.969 | 0.626 | 0.596 | 0.858 | **0.520** | 0.564 | 0.859 | 0.647 | 0.527 |
| GEMBA-MQM[†] | 0.802 | **0.993** | 0.502 | 0.572 | **0.939** | 0.401 | 0.564 | **0.991** | 0.449 | 0.522 |
| XCOMET-QE-Ensemble[†] | **0.808** | 0.974 | **0.679** | 0.588 | 0.909 | 0.498 | 0.554 | 0.892 | 0.647 | 0.533 |
| CometKiwi-QE | 0.781 | 0.946 | 0.475 | 0.569 | 0.859 | 0.387 | 0.544 | 0.963 | 0.442 | 0.525 |
| MetricX-23-QE-L | 0.763 | 0.868 | 0.501 | 0.577 | 0.616 | 0.419 | 0.536 | 0.835 | 0.622 | 0.508 |
| MetricX-23-QE-XXL | 0.797 | 0.934 | 0.547 | 0.607 | 0.813 | 0.459 | 0.575 | 0.877 | 0.652 | 0.531 |
| CometKiwi-QE-XL | 0.786 | 0.976 | 0.447 | 0.571 | 0.900 | 0.384 | 0.533 | 0.974 | 0.430 | 0.522 |
| ENSEMBLE BASELINES | | | | | | | | | | |
| Uniform | 0.801 | 0.935 | 0.552 | 0.600 | 0.816 | 0.484 | 0.566 | 0.948 | 0.637 | **0.540** |
| Weighted Avg | 0.802 | 0.934 | 0.555 | 0.603 | 0.811 | 0.487 | 0.568 | 0.943 | 0.647 | **0.540** |
| METAMETRICS-MT-QE | | | | | | | | | | |
| GP | 0.801 | 0.934 | 0.556 | **0.609** | 0.815 | 0.474 | **0.578** | 0.900 | **0.660** | 0.537 |
| XGBoost | 0.805 | 0.967 | 0.583 | 0.604 | 0.881 | 0.509 | 0.568 | 0.869 | 0.642 | 0.526 |

**GP-ALL (TQ).** The selected metrics are `code-inst-following`, `helpsteer-coherence`, `helpsteer-complexity`, `helpsteer-verbosity`, `ultrafb-inst_following`, `ArmoRM-overall`, `meteor`, `rouge`, and `rougel`.

**GP-ALL (NarrativeQA).** The selected metrics are `beavertails-is_safe`, `code-readbility`, `code-style`, `ultrafb-inst_following`, `ultrafb-truthfulness`, `ArmoRM-overall`, `bertscore-f1`, `bertscore-p`, `bertscore-r`, `bleu4`, `meteor`, and `rouge`.

**GP-ALL (SemEval).** The selected metrics are `code-complexity`, `code-explanation`, `code-inst-following`, `code-readability`, `code-style`, `helpsteer-coherence`, `helpsteer-helpfulness`, `prometheus-score`, `ultrafb-overall`, `bertscore-r`, `meteor`, and `rougel`.

**GP-ALL (Combined).** The selected metrics are `code-style`, `helpsteer-correctness`, `helpsteer-verbosity`, `ArmoRM-overall`, `bleu4`, `meteor`, and `rouge`.

**GP-Top 5 (NQ).** The selected metrics are `bertscore-r`, `bleu1`, `meteor`, `rouge` and `rougel`.

**GP-Top 5 (TQ).** The selected metrics are `helpsteer-correctness`, `bleu1`, `meteor`, `rouge`, and `rougel`.

**GP-Top 5 (NarrativeQA).** The selected metrics are `bertscore-f1`, `bertscore-r`, `meteor`, `rouge`, and `rougel`.

Table 13: Kendall correlation results of METAMETRICS-CAP is tuned on THumB 1.0 and tested on Flickr8k, and vice versa. Scores are reported for multiple metrics, with the average performance (**avg.**) across datasets. **Bold** and underlined values indicate the best and second best performance, respectively.

| Metric | Flickr8k | THumB 1.0 | avg. |
|---|---|---|---|
| BLEU-1 | 0.377 | 0.264 | 0.321 |
| BLEU-4 | 0.361 | 0.210 | 0.286 |
| ROUGE | 0.375 | 0.237 | 0.306 |
| METEOR | 0.481 | 0.281 | 0.381 |
| CIDEr | 0.498 | 0.263 | 0.381 |
| SPICE-F | 0.591 | 0.200 | 0.396 |
| SPICE-PR | 0.603 | 0.170 | 0.387 |
| SPICE-RE | 0.579 | 0.221 | 0.400 |
| TIGEr | 0.542 | 0.153 | 0.348 |
| CLIP-S | 0.514 | 0.292 | 0.403 |
| RefCLIP-S | 0.576 | 0.210 | 0.393 |
| METAMETRICS-CAP | | | |
| GP (All) | 0.664 | **0.329** | **0.496** |
| GP (Top 5) | 0.655 | 0.277 | 0.466 |
| XGBoost (All) | **0.670** | 0.277 | 0.474 |
| XGBoost (Top 5) | 0.664 | 0.283 | 0.474 |
| XGBoost (Iterative-Best) | 0.662 | 0.285 | 0.474 |
| XGBoost (Iterative-Top5) | 0.659 | 0.276 | 0.468 |
| METAMETRICS-CAP (Cross-Dataset) | | | |
| GP (All) | 0.638 | 0.298 | 0.468 |
| GP (Top 5) | 0.560 | 0.246 | 0.403 |
| XGBoost (All) | 0.595 | 0.323 | 0.449 |
| XGBoost (Top 5) | 0.575 | 0.311 | 0.443 |
| XGBoost (Iterative-Best) | 0.635 | 0.321 | 0.478 |
| XGBoost (Iterative-Top 5) | 0.634 | 0.301 | 0.467 |

**GP-Top 5 (SemEval).**  The selected metrics are `helpsteer-coherence`, `bleu4`, `meteor`, `rouge`, and `rougel`.

**GP-Top 5 (Combined).**  The selected metrics are `argilla-judge_llm`, `bleu1`, `meteor`, `rouge`, and `rougel`.

## G.2  MACHINE TRANSLATION

There are 2 settings for MT: reference-based and reference-free. Both setups' resulting weights or feature importance are reported in Figure 6. The weights selected for GP are as follows:

**GP-ALL (Reference-based).**  The selected metrics are `MetricX-XXL`, `COMET`, and `XCOMET-XL`.

**GP-ALL (Reference-free).**  The selected metrics are `MetricX-QE-XXL`, `MetricX-QE-Large`, `COMETKiwi-QE`, and `COMETKiwi-XL-QE`.

## G.3  IMAGE CAPTIONING

We utilize two datasets for Image Captioning, each with two setups: a) cross-dataset and b) in-dataset. The resulting weights and feature importance are presented in Figure 7.

The weights selected in GP are as follows:

Table 14: Accuracy results for Reward-Model-As-A-Metric. **Bold** and underlined values indicate the best and second best performance, respectively.

| Metric | Score | Chat | Chat Hard | Safety | Reasoning |
|---|---|---|---|---|---|
| Llama-3.1-Nemotron-70B-Reward | **94.1** | 97.5 | 85.7 | **95.1** | 98.1 |
| Skywork-Reward-Gemma-2-27B | 93.8 | 95.8 | **91.4** | 91.9 | 96.1 |
| TextEval-Llama3.1-70B | 93.5 | 94.1 | 90.1 | 93.2 | 96.4 |
| Skywork-Critic-Llama-3.1-70B | 93.3 | 96.6 | 87.9 | 93.1 | 95.5 |
| URM-LLaMa-3.1-8B | 92.9 | 95.5 | 88.2 | 91.1 | 97.0 |
| SFR-LLaMa-3.1-70B-Judge-r | 92.7 | 96.9 | 84.8 | 91.6 | 97.6 |
| Skywork-Reward-Llama-3.1-8B | 92.5 | 95.8 | 87.3 | 90.8 | 96.2 |
| GRM-Llama3-8B | 91.5 | 95.5 | 86.2 | 90.8 | 93.6 |
| Internlm2-20B-Reward | 90.2 | 98.9 | 76.5 | 89.5 | 95.8 |
| ArmoRM-Llama3-8B-v0.1 | 90.4 | 96.9 | 76.8 | 90.5 | 97.3 |
| URM-LLaMa-3-8B | 89.9 | 96.9 | 78.7 | 88.2 | 95.7 |
| Internlm2-7B-Reward | 87.6 | **99.2** | 69.5 | 87.2 | 94.5 |
| ENSEMBLE BASELINES | | | | | |
| Uniform | 92.5 | 98.3 | 83.1 | 90.8 | 97.6 |
| Weighted Avg. | 92.5 | 98.3 | 83.1 | 90.7 | 97.6 |
| METAMETRICS-RM | | | | | |
| GP | 93.5 | 98.9 | 86.2 | 90.7 | **98.2** |
| GP (Top 3) | 92.8 | 98.3 | 84.4 | 90.7 | 97.9 |
| XGBoost | 92.9 | 95.8 | 89.7 | 92.2 | 94.0 |
| XGBoost (Iterative Best) | 92.7 | 95.5 | 89.0 | 91.6 | 94.7 |
| XGBoost (Iterative Top 3) | 92.3 | 95.5 | 87.7 | 91.2 | 94.8 |
| XGBoost (Top 3) | 92.2 | 96.6 | 86.2 | 91.2 | 94.7 |

Table 15: Ablation study on GP using linear, multiplicative, and combined (linear and multiplicative) weightings on Machine Translation (MT), and Reward Model tasks.

| Method | MT | MT (QE) | Reward Model |
|---|---|---|---|
| GP (Linear) | 0.819 | 0.801 | 93.5 |
| GP (Multiplicative) | 0.822 | 0.798 | 91.2 |
| GP (Combined) | 0.818 | 0.803 | 92.1 |

Table 16: Kendall correlation results with human ratings for summarization task with Spearman and Kendall as the evaluation functions. **Coh.**, **Cons.**, **Rel.**, and **Faith.** corresponds to coherence, consistency, relevance, and faithfulness respectively.

| Metric | SummEval | | | | Benchmark LLM | | | Combined | | |
|---|---|---|---|---|---|---|---|---|---|---|
| | Coh. | Cons. | Fluency | Rel. | Faith. | Coh. | Cons. | Coh. | Cons. | Avg. |
| XGBoost (Kendall) | 0.475 | **0.376** | **0.407** | 0.439 | 0.406 | 0.411 | 0.397 | **0.403** | 0.367 | 0.409 |
| XGBoost (Spearman) | **0.476** | **0.376** | 0.403 | **0.448** | 0.406 | 0.411 | 0.397 | 0.402 | **0.369** | 0.409 |

**GP-ALL (Flickr8k).** The selected metrics are `cider`, `clipscore`, `meteor`, `ref-clipscore`, `spice_f`, `spice_pr`, `spice_re`, and `tiger`.

**GP-ALL (THumB 1.0).** The selected metrics are `bleu1`, `clipscore`, `meteor`, `ref-clipscore`, `rouge`, `spice_re`, and `tiger`.

**GP Top 5 (Flickr8k).** The selected metrics are `ref-clipscore`, `spice_f`, `spice_pr`, `spice_re`, and `tiger`.

Table 17: Results on Cross-lingual settings on WMT23 for Hebrew-English (he-en). **Bold** and underline values indicate the best and second best performance, respectively.

| Metric | he-en | | |
| --- | --- | --- | --- |
| | sys pearson | seg pearson | seg acc-t |
| chrF[†] | 0.776 | 0.221 | 0.460 |
| BLEU[†] | 0.769 | 0.220 | 0.442 |
| BERTScore[†] | 0.895 | 0.335 | 0.515 |
| Yisi-1[†] | 0.917 | 0.395 | 0.529 |
| MetricX-23-XXL[†] | 0.910 | 0.548 | 0.577 |
| XCOMET-Ensemble[†] | 0.950 | 0.556 | 0.586 |
| COMET | 0.940 | 0.401 | 0.531 |
| XCOMET-XL | 0.949 | 0.530 | 0.576 |
| XCOMET-XXL | 0.952 | 0.465 | 0.560 |
| METAMETRICS-MT | | | |
| GP | 0.947 | 0.546 | **0.590** |
| XGBoost | **0.957** | **0.557** | 0.574 |

Table 18: Average Results of acc-t and Soft-pairwise accuracy (SPA) on Cross-lingual settings on WMT24 (Freitag et al., 2024) for English-Spanish (en-es) and Japanese-Chinese (ja-zh). We report the number following the metric calculated by WMT24 organizers for language breakdown performance since the test set labels are unseen to the authors. **Bold** and underline values indicate the best and second best performance, respectively.

| Metric | en-es | ja-zh | avg |
| --- | --- | --- | --- |
| sentinel-ref-mqm | 0.631 | 0.490 | 0.561 |
| BLEU | 0.596 | 0.588 | 0.592 |
| spBLEU | 0.602 | 0.590 | 0.596 |
| chrF | 0.615 | 0.616 | 0.616 |
| chrfS | 0.630 | 0.602 | 0.616 |
| BERTScore | 0.594 | 0.651 | 0.623 |
| MEE4 | 0.635 | 0.625 | 0.630 |
| damonmonli | 0.688 | 0.633 | 0.661 |
| YiSi-1 | 0.657 | 0.666 | 0.662 |
| PrismRefSmall | 0.666 | 0.667 | 0.667 |
| PrismRefMedium | 0.734 | 0.545 | 0.640 |
| BLCOM_1 | 0.698 | 0.676 | 0.687 |
| BLEURT-20 | 0.688 | 0.685 | 0.687 |
| COMET-22 | 0.744 | 0.636 | 0.690 |
| XCOMET | 0.740 | 0.700 | 0.720 |
| MetricX-24 (Hybrid) | 0.742 | **0.718** | 0.730 |
| METAMETRICS-MT (GP) | **0.745** | 0.717 | **0.731** |

**GP Top 5 (THumB 1.0).**   The selected metrics are `bleu1`, `cider`, `clipscore`, `meteor`, and `rouge`.

## G.4   TEXT SUMMARIZATION

The weights and feature importance from the resulting optimization are reported in Figure 8, Figure 9, and Figure 10 for SummEval, BLLM, and the merged dataset respectively. The weights selected in GP are as follows:

### G.4.1   W/O LLM-BASED METRICS

**GP-ALL (SummEval:   Coherence).**   The selected metrics are `BLEURT (Max)`, `chrF`, `Rouge-1`, `Rouge-4`, and `SummaQA (Prob)`.

Table 19: Kendall correlation Cross-Dataset results of METAMETRICS-CAP when it is tuned on THumB 1.0 and tested on Flickr8k, and vice versa. The columns correspond to the testing dataset. Scores are reported for multiple metrics, with the average performance (**avg.**) across Cross-Dataset results. **Bold** and underlined values indicate the best and second best performance, respectively.

| Metric | Flickr8k | THumB 1.0 | avg. |
|---|---|---|---|
| BLEU-1 | 0.377 | 0.264 | 0.321 |
| BLEU-4 | 0.361 | 0.210 | 0.286 |
| ROUGE | 0.375 | 0.237 | 0.306 |
| METEOR | 0.481 | 0.281 | 0.381 |
| CIDEr | 0.498 | 0.263 | 0.381 |
| SPICE-F | 0.591 | 0.200 | 0.396 |
| SPICE-PR | 0.603 | 0.170 | 0.387 |
| SPICE-RE | 0.579 | 0.221 | 0.400 |
| TIGEr | 0.542 | 0.153 | 0.348 |
| CLIP-S | 0.514 | 0.292 | 0.403 |
| RefCLIP-S | 0.576 | 0.210 | 0.393 |
| METAMETRICS-CAP | | | |
| GP (All) | **0.638** | 0.298 | 0.468 |
| GP (Top 5) | 0.560 | 0.246 | 0.403 |
| XGBoost (All) | 0.595 | **0.323** | 0.449 |
| XGBoost (Top 5) | 0.575 | 0.311 | 0.443 |
| XGBoost (Iterative-Best) | 0.635 | 0.321 | **0.478** |
| XGBoost (Iterative-Top 5) | 0.634 | 0.301 | 0.467 |

**GP-ALL (SummEval: Consistency).** The selected metrics are `BERTScore (Recall)`, `BLEURT (Max)`, `chrF`, `SummaQA (f1)`, and `SummaQA (Prob)`.

**GP-ALL (SummEval: Fluency).** The selected metrics are `BERTScore (Recall)`, `BLEURT (Max)`, `chrF`, `SummaQA (f1)`, and `SummaQA (Prob)`.

**GP-ALL (SummEval: Relevance).** The selected metrics are `BERTScore (Recall)`, `BLEURT (Mean)`, `chrF`, `Rouge-WE1`, `Rouge-1`, `Rouge-4`, and `SummaQA (Prob)`.

### G.4.2   W/ LLM-BASED METRICS

**GP-ALL (SummEval: Coherence).** The selected metrics are `BARTScore (Mean)`, `BLEURT (Max)`, `BLEURT (Mean)`, `Rouge-3`, `Rouge-4`, `SummaQA (Prob)`, `G-Eval`, and `UniEval`.

**GP-ALL (SummEval: Consistency).** The selected metrics are `BLEURT (Max)`, `SummaQA (Prob)`, `G-Eval`, and `UniEval`.

**GP-ALL (SummEval: Fluency).** The selected metric is `G-Eval`.

**GP-ALL (SummEval: Relevance).** The selected metrics are `BLEURT (Mean)`, `chrF`, `Rouge-3`, `Rouge-4`, `SummaQA (Prob)`, `G-Eval`, and `UniEval`.

**GP-Top 2 (SummEval: Coherence).** The selected metrics are `G-Eval` and `UniEval`.

**GP-Top 2 (SummEval: Consistency).** The selected metrics are `G-Eval` and `UniEval`.

**GP-Top 2 (SummEval: Fluency).** The selected metrics are `G-Eval` and `UniEval`.

**GP-Top 2 (SummEval: Relevance).** The selected metrics are `G-Eval` and `UniEval`.

Table 20: Runtime for inference time on train and test split and the calibration time for METAMETRICS. Rows with exact same run time is due to the groups of metrics (e.g., BLEURT (Max) and BLEURT (Mean)) being calculated in a single pass. †G-Eval (GPT4) costs around US$82 to run SummEval benchmark with a max token of 5, temperature of 2, top p of 1, a frequency penalty of 0, a presence penalty of 0, and completion choices of 20. The running time reported for METAMETRICS-SUM for train and test split is in a sequential manner. In other words, running each metric in an embarrassingly parallel way can reduce the overall inference time. To compare the proper time evaluation on test split with METAMETRICS-SUM, it is important to account for all the components of METAMETRICS-SUM's evaluation process. Specifically, for individual metrics, we can consider the inference time on the test split. On the other hand, for METAMETRICS-SUM, we need to consider the inference time on both the train and test splits, and then the calibration time using the train set. For example, G-Eval (GPT4) takes 7,173.50 seconds to evaluate on the test split, while XGBoost (all metrics including LLM-based metrics) takes 3,976.47 + 9,292.83 + 647.08 = 13,916.38 seconds to evaluate on the test split.

| Metric | Inference Time (in sec) | | Calibration Time (in sec) |
|---|---|---|---|
| | Train Split | Test Split | |
| BLEU | 0.72 | 1.73 | N/A |
| chrF | 3.37 | 8.67 | N/A |
| METEOR | 5.74 | 13.57 | N/A |
| ROUGE-1 | 57.19 | 162.77 | N/A |
| ROUGE-3 | 68.24 | 151.90 | N/A |
| ROUGE-4 | 70.43 | 149.65 | N/A |
| ROUGE-l | 66.53 | 155.15 | N/A |
| ROUGE-WE1 | 56.35 | 100.19 | N/A |
| BLEURT (Max) | 139.87 | 336.22 | N/A |
| BLEURT (Mean) | 139.87 | 336.22 | N/A |
| BERTScore (f1) | 1.74 | 3.99 | N/A |
| BERTScore (Recall) | 1.74 | 3.99 | N/A |
| SummaQA (f1) | 322.07 | 790.11 | N/A |
| SummaQA (Prob) | 322.07 | 790.11 | N/A |
| LLM-BASED METRICS | | | |
| BARTScore (Max) | 22.39 | 50.83 | N/A |
| BARTScore (Mean) | 22.39 | 50.83 | N/A |
| UniEval | 80.37 | 194.55 | N/A |
| G-Eval (GPT4) | 3,081.46† | 7,173.50† | N/A |
| ENSEMBLE BASELINES | | | |
| Uniform | 3,976.47† | 9,292.83† | N/A |
| Weighted Avg | 3,976.47† | 9,292.83† | N/A |
| METAMETRICS-SUM | | | |
| GP | 792.25 | 1,873.95 | 402.51 |
| XGBoost | 792.25 | 1,873.95 | 276.52 |
| W/ LLM-BASED METRICS | | | |
| GP | 3,976.47† | 9,292.83† | 647.08 |
| GP (Top 2) | 3,161.83† | 7,368.05† | 187.73 |
| XGBoost | 3,976.47† | 9,292.83† | 337.88 |
| XGBoost (Top 2) | 3,161.83† | 7,368.05† | 223.69 |

**GP-ALL (BLLM: Faithfulness).** The selected metrics are `chrF`, `SummaQA (f1)`, and `SummaQA (Prob)`.

**GP-ALL (BLLM: Coherence).** The selected metrics are `BARTScore`, `BERTScore (f1)`, `BLEURT`, `Rouge-1`, and `Rouge-L`.

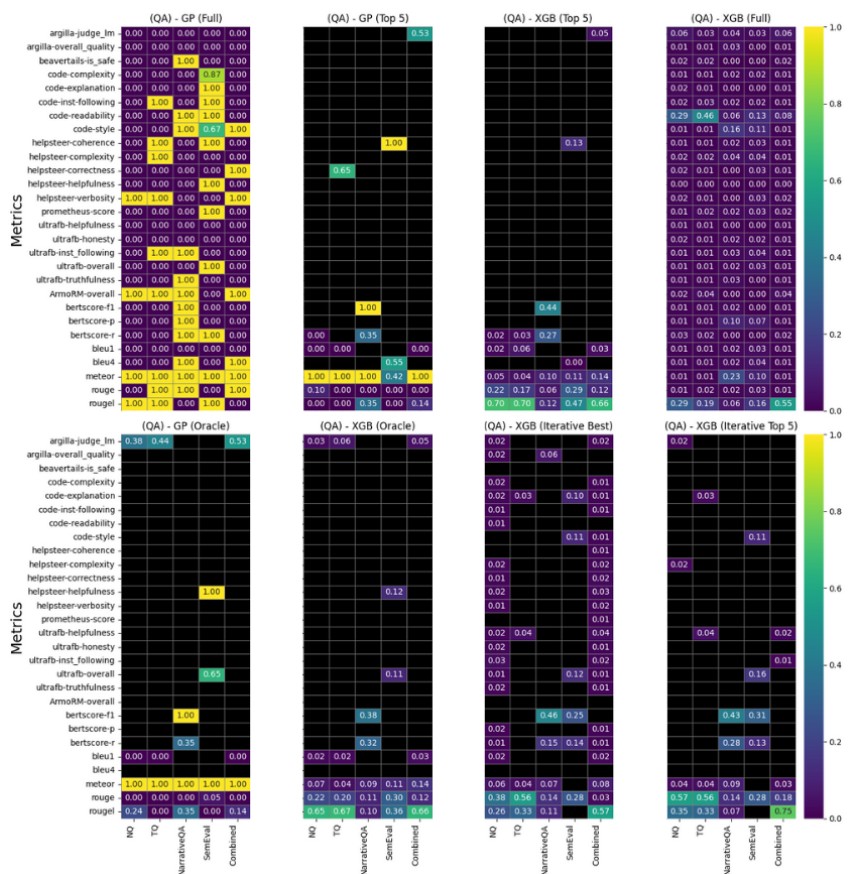

Figure 5: Weights and Feature Importance for METAMETRICS-QA.

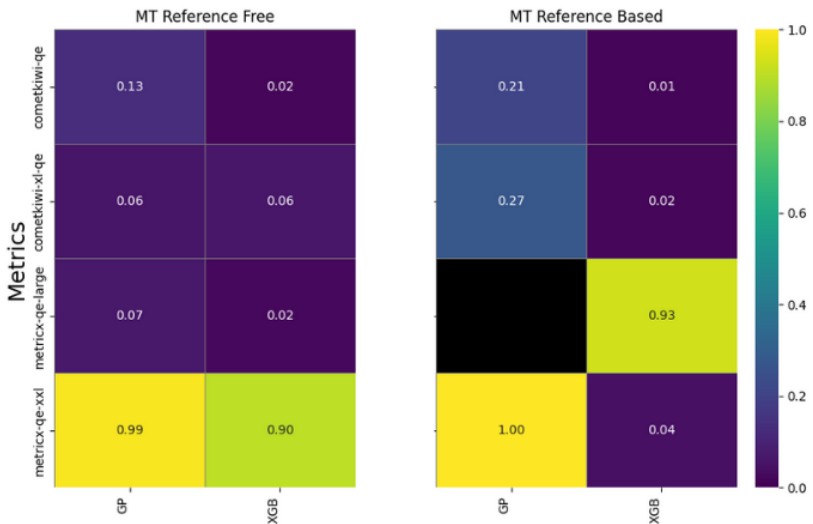

Figure 6: Weights and Feature Importance for METAMETRICS-MT.

**GP-ALL (BLLM: Relevance).** The selected metrics are `BERTScore (Recall)`, `chrF`, `METEOR`, `Rouge-WE1`, `SummaQA (f1)`, and `SummaQA (Prob)`.

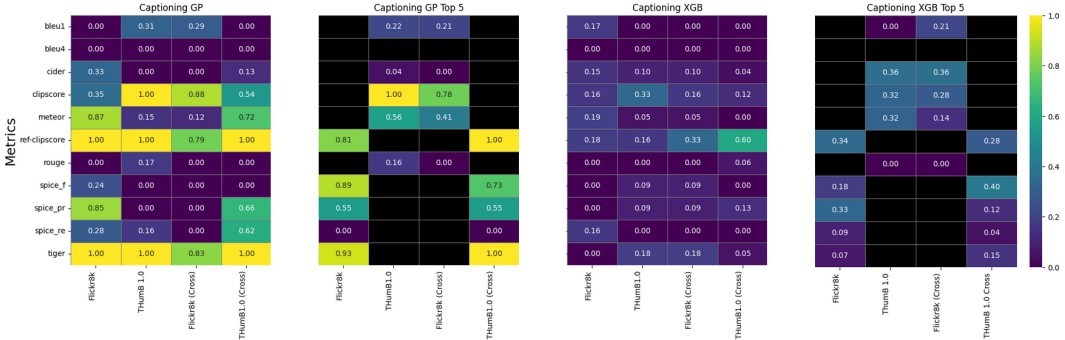

Figure 7: Weights and Feature Importance for METAMETRICS-CAP.

**GP-Top 5 (BLLM: Faithfulness).** The selected metrics are `METEOR`, `ROUGE-WE1`, `SummaQA (f1)` and `SummaQA (Prob)`.

**GP-Top 5 (BLLM: Coherence).** The selected metrics are `BERTScore (f1)`, `BLEURT`, `Rouge-1`, and `Rouge-L`.

**GP-Top 5 (BLLM: Relevance).** The selected metrics are `BERTScore (Recall)`, `chrF`, `METEOR`, `Rouge-WE1`, and `SummaQA (Prob)`.

**GP-ALL (Combined: Coherence).** The selected metrics are `BARTScore (Mean)`, `BERTScore (f1)`, and `BLEURT (Mean)`.

**GP-ALL (Combined: Relevance).** The selected metrics are `BERTScore (f1)`, `BERTScore (Recall)`, `BLEURT (Mean)`, `SummaQA (f1)` and `SummaQA (Prob)`.

**GP-Top 5 (Combined: Coherence).** The selected metrics are `BERTScore (f1)` and `BLEURT (Mean)`.

**GP-Top 5 (Combined: Relevance).** The selected metrics are `BERTScore (f1)`, `BERTScore (Recall)`, `Rouge-WE1`, and `Rouge-1`.

### G.5 REWARD MODELING

The weights and feature importance from the resulting optimization are reported in Figure 11. The weights selected in GP are as follows:

**GP-ALL.** The selected metrics are `GRM-Gemma-2B-rewardmodel-ft_all`, `GRM-Llama3-8B-rewardmodel-ft_all`, `Skyword-Reward-Llama-3.1-8B_all`, `URM-LLaMa-3.1-8B_all`, `internlm2-1_8b-reward_all`, and `internlm2-7b-reward_all`.

**GP-Top 3.** The selected metrics are `GRM-Llama3-8B-rewardmodel-ft_all`, `Skyword-Reward-Llama-3.1-8B_all`, and `internlm2-7b-reward_all`.

## H CORRELATION MEASUREMENT DETAILS

This section describes the details of correlation measurements used in our experiments.

**Kendall's Tau** Kendall's Tau measures rank correlation by assessing the agreement between two rankings based on the proportion of concordant and discordant pairs. It is commonly used to evaluate how closely an evaluation metric's rankings align with human rankings in tasks like Machine Translation.

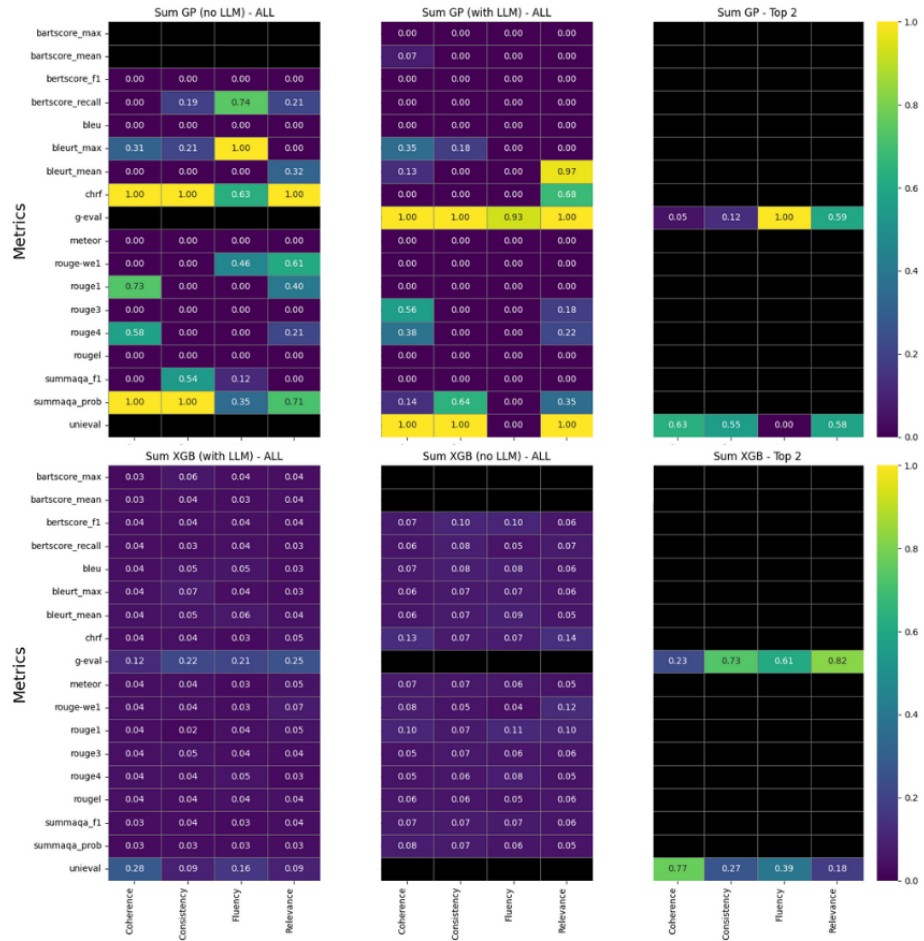

Figure 8: Weights and Feature Importance for METAMETRICS-SUMM on the SummEval dataset.

## H.1 MACHINE TRANSLATION TASK CORRELATION

**System-Level Pearson Correlation (sys pearson)** This metric evaluates the correlation between metric and human scores aggregated at the system level. It is calculated as the Pearson correlation coefficient between the averaged scores of each system.

**Segment-Level Pearson Correlation (seg pearson)** At the segment level, correlation is computed by flattening the system × segment score matrices into vectors and comparing the metric-generated vector with the human-evaluation vector. This provides a more granular assessment of alignment.

**System-level pairwise ranking accuracy (seg acc-t)** System-level pairwise ranking accuracy, as proposed by Kocmi et al. (2021) measures the agreement between a metric and human judgments in ranking translation systems. It is computed as the percentage of system pairs for which the metric's ranking matches human-provided rankings. Rankings are based on pooled data across all language pairs, making this metric robust for system comparisons.

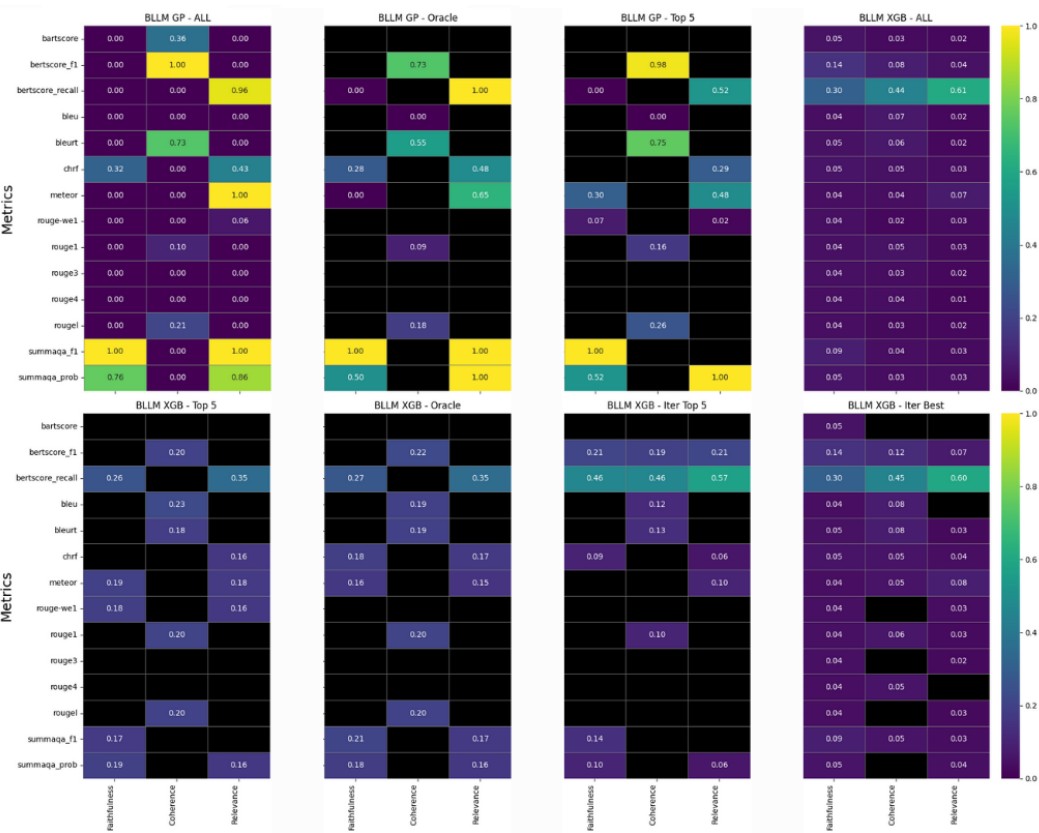

Figure 9: Weights and Feature Importance for METAMETRICS-SUMM on the BLLM dataset.

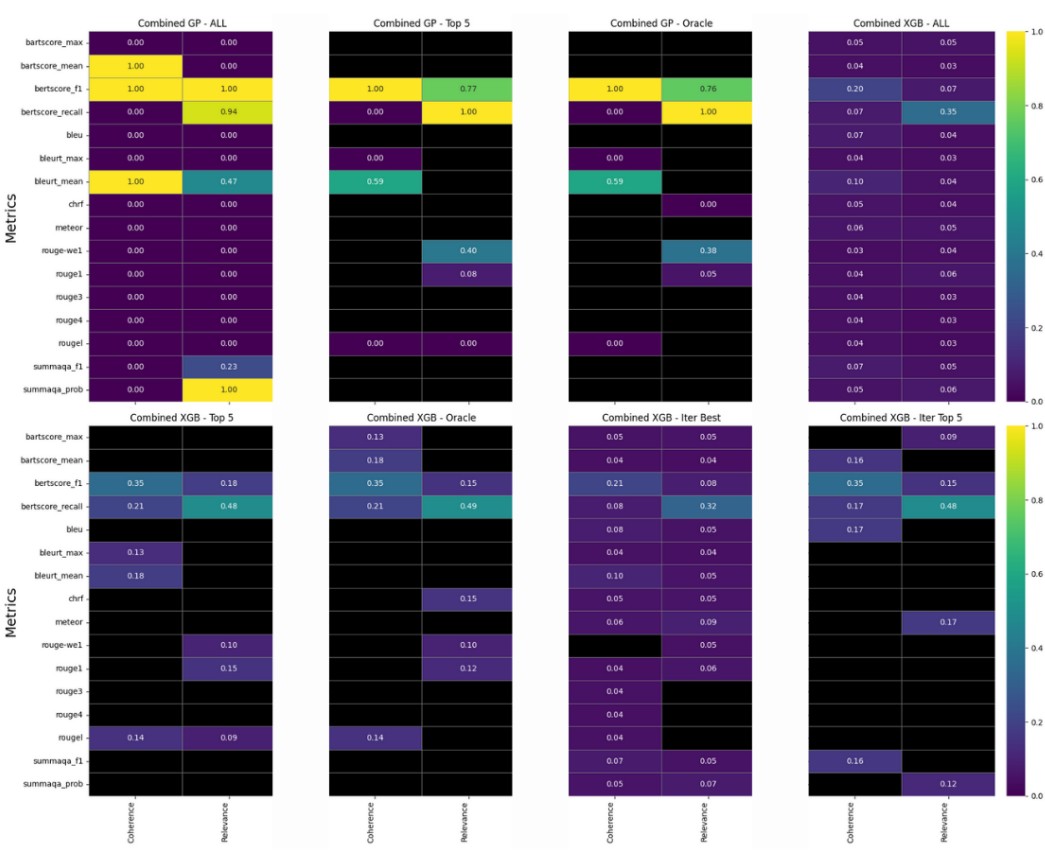

Figure 10: Weights and Feature Importance for METAMETRICS-SUMM on the merged dataset.

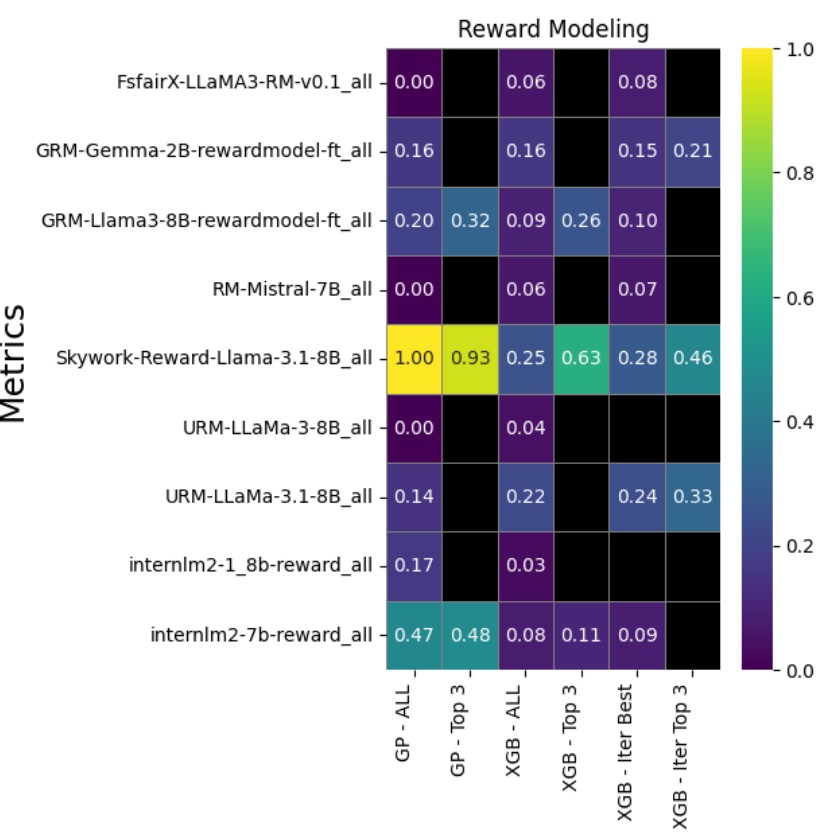

Figure 11: Weights and Feature Importance for METAMETRICS-RM.

