# OpenReview forum: "MetaMetrics: Calibrating Metrics for Generation Tasks Using Human Preferences"
_ICLR.cc/2025/Conference — ICLR 2025 Poster_

### Official Review · Reviewer_nKfe · 2024-11-02

**Soundness:** 2
**Presentation:** 2
**Contribution:** 2
**Rating:** 5
**Confidence:** 4

**Summary:**

This paper introduces a novel evaluation metric, MetaMetric, designed to better align with human preferences. The proposed metric is a weighted combination of existing metrics, where the weights are learned in a supervised manner. The authors demonstrate the effectiveness of MetaMetric through experiments on various natural language generation tasks, e.g., summarization, translation, question answering and image captioning. Additionally, they show that this metric can be useful for reward model scoring. They also provide some analysis about interpretability and efficiency of their metric.

**Strengths:**

1. The experiments are comprehensive, including various types of natural language generation tasks.
2. The topic of developing a human-aligned evaluation metric for NLP generation tasks is a long-discussed but unsolved challenge. This paper explores a possibility to push forward in this direction.
3. The proposed metric is easily extendable to other tasks where preliminary metrics have already existed. This extensibility makes the metric potentially applicable to a broader range of applications.

**Weaknesses:**

I have several concerns regarding the experiments, particularly the scope of the problem and the evaluation settings, and how they support the authors' claims. Additionally, the presentation of the paper requires refinement.
1. **Limited Contribution:** The paper's contribution appears limited. It essentially addresses a traditional supervised learning problem with input of small dimensions and output with single normalized value. The authors train a regression model on supervised-learning datasets, using conventional metrics as inputs and human preference values as outputs. The experimental results involve testing this regression model on the test set of these datasets and comparing different metrics using correlation of the rankings of metrics with ground truth human preference values. It is unsurprising that MetaMetric outperforms other metrics in the in-domain generalization setting, which is the primary focus of this paper. The authors should also clearly present their experimental settings, such as how they create the training, validation and testing sets. For example, the setting for Question Answering tasks is not clear. It would be interesting to add more OOD and real scenarios to test this metric. See following comments for detailed feedback.
2. **Lack of Out-of-domain (OOD) Generalization Evaluation:** The authors should place more emphasis on the OOD setting, as human preference data is not always available. I found only one one OOD setting presented in Figure 3. Based on in-domain performance, the GP model works the best. However, the GP model performs similarly to CLIP-S and METEOR in the OOD setting. (The statement about the superiority of MetaMetric-Cap over any other individual metric, in lines 426 and 427, is misleading.) This suggests that the use cases for the trained metric are limited, and its performance may be worse than using a single conventional metric. It would al os be interesting to evaluate metaMetric trained on one task but tested on another, which is a more challenging OOD setting than the one in Figure 3.
3. **Lack of Evaluation in Real Settings:** The evaluations are limited by the training datasets with human-annotated preference values. It is unclear whether this metric is useful for real applications, as suggested by the authors in lines 41 and 42. For example, the authors could follow a similar evaluation procedure to DPO and PPO papers, using the proposed metric to evaluate various LLMs - before and after RLHF-tuned - and compare their win rates based on MetaMetric.
4. **Reference-based and Reference-free Settings:** In line 83, the authors claim to design the metric for two settings: reference-based and reference-free. If I understand correctly, this depends on the choice of metrics rather than the model design. If the chosen metrics require references, then MetaMetric can only be reference-based. In the experiments, the authors only distinguish these two settings for Machine Translation, not for other tasks. It would be better that the authors add both settings for all the tasks.
5. **Presentation Issues:** The paper's presentation is suboptimal. Major issues include:
    1. The caption of a table should be placed above it.
    2. Figure 2 is not cited anywhere in the paper.
    3. The authors cite a table in the Appendix as crucial support for their claims (see line 357 for Table 9). The authors should put the information in the main text.
    4. Other minor issues are listed in the Questions section.

**Questions:**

1. **Questions related to Introduction Section:**
    1. The first paragraph is a bit confusing. The authors discuss the shortcomings of BLEU and BERTScore in the main text, but Figure 1 presents BLEU and CLIP-S. Additionally, what criteria are used in Figure 1 to determine if a metric is good or not? For the figure on the right hand side, why are BLEU 0.2 and MeMetric 0.4 considered good, but CLIP-S 0.6 is considered bad? What is the threshold?
    2. The authors state, "models that optimize for human preference ... have demonstrated superior performance." What metrics have these models used to demonstrate human alignment? The authors avoid discussing this in their paper, instead comparing their metric with conventional metrics not originally designed for human preference alignment. For example, Ouyang et al. [1] used the Perspective AI score to measure toxicity.
    3. In line 42, the authors mention "evaluation metrics that accurately reflect human subjective judgements." Since human judgements are **subjective**, is it possible that a metric can **accurately** measure these varied judgements across different individuals?
    4. In line 53, the statement relating the necessity of human aligned metrics to the complexity of tasks seems implausible. Regardless of task complexity, we always seek metrics that align with human evaluations.
2. **Universality of MetaMetric:** For conventional metrics (e.g., BLEU, ROUGE, METEOR), they are used across various natural language generation tasks. Is it feasible to have a single MetaMetric applicable to all natural language generation tasks?
3. **Choice of Training Methods:** Why did the authors choose Bayesian Optimization and Boosting to train MetaMetrics? Is it possible to use linear regression models with regularizations?
4. **Efficiency of MetaMetric:**. According to Figure 4, MetaMetric requires about 8 metrics for computation. Can the authors explain why computing approximately 8 metrics is still considered efficient?
5. **Interpretability:** Can the authors explain why some metrics align more closely with human evaluations while others do not? I'm curious whether there is any intuition behind this.
6. **Compute Efficiency for RM:** In lines 433 and 472, Table 4 does not show compute efficiency. The authors should include the model size for MetaMetrics-RM.

[1] Ouyang, Long, et al. "Training language models to follow instructions with human feedback." Advances in neural information processing systems 35 (2022): 27730-27744.

**Minor Issues:**
1. In line 215, there is a duplicate "is"
2. Presentation suggestion: Section 2 could be merged with Section 7.

---

> ### Author Response · Authors · 2024-11-20
> **Updated Paper, Addressing Concerns, and Clarification Part 1**
>
> Thank you for your thorough comments. We have addressed your questions and concerns as follows:
>
> > Limited Contribution and Lack of Out-of-domain (OOD) Generalization Evaluation
>
> We have presented results for both **in-domain and out-of-domain settings**, specifically highlighting the robustness and generalization capabilities of our methods in Machine Translation, Image Captioning, and Reward Modeling. In Machine Translation, we demonstrate that our metric can generalize to unseen languages, such as Hebrew-to-English (he-en). Additionally, we illustrate the **cross-domain and cross-dataset adaptability** of our metric in the areas of image captioning and reward modeling. For a detailed overview, please refer to Tables 2 (Machine Translation) and 4 (Reward Modeling), as well as Figure 2 (Image Captioning), which emphasize these findings.
>
> > The authors should also clearly present their experimental settings, such as how they create the training, validation and testing sets. For example, the setting for Question Answering tasks is not clear. It would be interesting to add more OOD and real scenarios to test this metric. See following comments for detailed feedback.
>
> We have included comprehensive details for running the experiments, including **task-specific splits**, in Section D (Datasets) of the Appendix. To enhance clarity, we have also specified the **30%-70% train-test split ratio** in Section 4 (Experimental Setup) for your convenience.
>
> > GP model performs similarly to CLIP-S and METEOR in the OOD setting. (The statement about the superiority of MetaMetric-Cap over any other individual metric, in lines 426 and 427, is misleading.) This suggests that the use cases for the trained metric are limited, and its performance may be worse than using a single conventional metric.
>
> We would like to clarify that our metric MetaMetrics-Cap outperforms all other individual metrics in Figure 2 (formerly Figure 3 prior to revision). This figure illustrates performance when trained on one dataset and tested on a different, out-of-domain (OOD) dataset. Detailed results are presented in Table 13 (formerly Table 12), where MetaMetrics-Cap (GP) achieves a score of 0.638, compared to CLIP-S at 0.514 and METEOR at 0.481 on the Flickr8k out-of-domain dataset. A similar trend is shown on Thumb1.0, where MetaMetrics-Cap (GP) achieves 0.298, surpassing CLIP-S at 0.292 and METEOR at 0.281, also in an out-of-domain setting.
>
> > Lack of Evaluation in Real Settings: The evaluations are limited by the training datasets with human-annotated preference values. It is unclear whether this metric is useful for real applications, as suggested by the authors in lines 41 and 42. For example, the authors could follow a similar evaluation procedure to DPO and PPO papers, using the proposed metric to evaluate various LLMs - before and after RLHF-tuned - and compare their win rates based on MetaMetric.
>
> Our experiment was conducted in a low-resource setting, **requiring only a minimal number of samples** during calibration. Specifically, we utilized 30% of the dataset for training while reserving the remaining 70% for testing. We believe this experimental setup effectively simulates real-world scenarios where human-annotated preference **data is scarce**, demonstrating that only a small portion of datasets needs to be human-annotated.
>
> > Reference-based and Reference-free Settings: In line 83, the authors claim to design the metric for two settings: reference-based and reference-free. If I understand correctly, this depends on the choice of metrics rather than the model design. If the chosen metrics require references, then MetaMetric can only be reference-based. In the experiments, the authors only distinguish these two settings for Machine Translation, not for other tasks. It would be better that the authors add both settings for all the tasks.
>
> We would like to clarify that not all tasks utilize reference-free metrics. In particular, for the machine translation task, reference-free settings are commonly used for evaluation. In contrast, other downstream tasks primarily rely on reference-based metrics. Therefore, our reference-free setting is exclusively applied to machine translation.
>
> > Presentation Issues
>
> As per your suggestion, we have moved all table captions to appear above their respective tables. Additionally, we have **moved Figure 2 to the Appendix**, as it serves a supplementary role, and have updated the text accordingly. The previous reference to Table 9 was an error; it has been corrected to refer to Table 1, where the relevant data is actually presented.

---

> ### Author Response · Authors · 2024-11-20
> **Updated Paper, Addressing Concerns, and Clarification Part 2**
>
> > The first paragraph is a bit confusing. The authors discuss the shortcomings of BLEU and BERTScore in the main text, but Figure 1 presents BLEU and CLIP-S. Additionally, what criteria are used in Figure 1 to determine if a metric is good or not? For the figure on the right hand side, why are BLEU 0.2 and MeMetric 0.4 considered good, but CLIP-S 0.6 is considered bad? What is the threshold?
>
> The figure compares BLEU and CLIP-S to illustrate **metric alignment with human judgments**, not to advocate for specific metrics. A "good" metric aligns closely with human ratings, scoring below 0.5 for low-quality instances and above 0.5 for high-quality ones. The apparent discrepancy (e.g., BLEU 0.2 and MetaMetrics 0.4 as "good" vs. CLIP-S 0.6 as "bad") reflects differences in alignment with human ratings.
>
> > The authors state, "models that optimize for human preference ... have demonstrated superior performance." What metrics have these models used to demonstrate human alignment? The authors avoid discussing this in their paper, instead comparing their metric with conventional metrics not originally designed for human preference alignment. For example, Ouyang et al. [1] used the Perspective AI score to measure toxicity.
>
> We wish to clarify that the statement in question was made based on the references provided. We added the statement according to the findings on the references. We didn’t intend to avoid any discussion regarding this matter. Let us know if you have further questions or clarification, we are happy to answer your question.
>
> > In line 42, the authors mention "evaluation metrics that accurately reflect human subjective judgements." Since human judgements are subjective, is it possible that a metric can accurately measure these varied judgements across different individuals?
>
> Automatic evaluation metrics serve as a proxy for human evaluation, which can be costly and time-consuming. Therefore, using automatic evaluation metrics is essential to offset the expenses and time associated with collecting human evaluations.
>
> > In line 53, the statement relating the necessity of human aligned metrics to the complexity of tasks seems implausible.
> Regardless of task complexity, we always seek metrics that align with human evaluations.
>
> Thank you for bringing this to our attention. We wish to clarify that our intent was to highlight the absolute necessity of human-aligned metrics in navigating the intricate, multi-aspects of human preference inherent in complex downstream tasks. This should not be misconstrued as implying that such metrics are exclusively requisite for only the more sophisticated or challenging tasks. Rather, we concur with the notion that human-aligned metrics carry significant importance across the entire spectrum of downstream tasks, irrespective of their respective levels of difficulty or complexity.
>
> > Universality of MetaMetric: For conventional metrics (e.g., BLEU, ROUGE, METEOR), they are used across various natural language generation tasks. Is it feasible to have a single MetaMetric applicable to all natural language generation tasks?
>
> We believe that even conventional metrics have advantages and disadvantages on each task. For example, BLEU is not well-correlated to human preferences in machine translation[1].
>
> [1] Freitag, M., Rei, R., Mathur, N., Lo, C. K., Stewart, C., Avramidis, E., ... & Martins, A. F. (2022, December). Results of WMT22 metrics shared task: Stop using BLEU–neural metrics are better and more robust. In Proceedings of the Seventh Conference on Machine Translation (WMT) (pp. 46-68).
>
> > Choice of Training Methods: Why did the authors choose Bayesian Optimization and Boosting to train MetaMetrics?
>
> In this work, we focus on two optimization methodologies to train MetaMetrics: Bayesian Optimization (BO) and Boosting. BO offers the advantage of **interpretability**, allowing us to clearly identify which metrics contribute most significantly to the final outcome. Conversely, the Boosting method excels in **enhancing alignment** and accounting for the **compositionality** of different metrics, even when dealing with more complex functions. Although we can measure the contribution of each metric, the clarity and distinctness of these contributions are more pronounced with BO compared to Boosting. Certainly, there are many alternative regressors that can be used as substitutes.

---

> > ### Author Response · Authors · 2024-11-20
> > **Updated Paper, Addressing Concerns, and Clarification Part 3**
> >
> > > Efficiency of MetaMetric:. According to Figure 4, MetaMetric requires about 8 metrics for computation. Can the authors explain why computing approximately 8 metrics is still considered efficient?
> >
> > In the figure displaying the QA results, we didn’t have any MetaMetrics utilizing 8 metrics; however, we do have MetaMetrics that employs 7 metrics. Concerning metric efficiency, some metrics, like BLEU, METEOR, and ROUGE, offer relatively quick inference times. While LLM-based metrics do require longer inference times, the overall inference duration remains minimal. To enhance efficiency further, we present results for XGBoost using 5 metrics, which outperforms the GP model using 7 metrics in the Combined QA setting. This approach not only boosts efficiency but also increases the metric's effectiveness.
> >
> > > Interpretability: Can the authors explain why some metrics align more closely with human evaluations while others do not? I'm curious whether there is any intuition behind this.
> >
> > Some metrics are designed to pick up on details that are similar to what humans notice, like how relevant something is in context, its meaning, and how natural it sounds. For example, ROUGE looks at overlapping words or sequences of words, which might not always capture the deeper meaning, but it works well in tasks like summarization where matching exact words is important.
> >
> > > Compute Efficiency and List of Base Metrics Used for MetaMetrics
> >
> > We added the list of models we use for MetaMetrics-RM on Appendix Section G.

---

> ### Author Response · Authors · 2024-11-24
> **Improved Robustness Section and Added New OOD Results**
>
> To address your concerns comprehensively, we have expanded the details in our robustness section (Section 6.3) and Appendix F.7. We provide an in-depth discussion on our observations and results in both cross-lingual and cross-dataset settings. To further substantiate our claims, we have included new results for the WMT24 cross-lingual evaluation on the English-Spanish (en-es) and Japanese-Chinese (jp-zh) language pairs.
>
> | Metric                | en-es | ja-zh | avg   |
> |-----------------------|-------|-------|------ |
> | sentinel-ref-mqm      | 0.631 | 0.490 | 0.561 |
> | BLEU                  | 0.596 | 0.588 | 0.592 |
> | spBLEU                | 0.602 | 0.590 | 0.596 |
> | chrF                  | 0.615 | 0.616 | 0.616 |
> | chrfS                 | 0.630 | 0.602 | 0.616 |
> | BERTScore             | 0.594 | 0.651 | 0.623 |
> | MEE4                  | 0.635 | 0.625 | 0.630 |
> | damonmonli            | 0.688 | 0.633 | 0.661 |
> | YiSi-1                | 0.657 | 0.666 | 0.662 |
> | PrismRefSmall         | 0.666 | 0.667 | 0.667 |
> | PrismRefMedium        | 0.734 | 0.545 | 0.640 |
> | BLCOM_1               | 0.698 | 0.676 | 0.687 |
> | BLEURT-20             | 0.688 | 0.685 | 0.687 |
> | COMET-22              | 0.744 | 0.636 | 0.690 |
> | XCOMET                | 0.740 | 0.700 | 0.720 |
> | MetricX-24 (Hybrid)   | 0.742 | **0.718** | 0.730 |
> | MetaMetrics-MT (GP)  | **0.745** | 0.717 | **0.731** |
>
> Additionally, for your convenience, we have summarized the cross-dataset results on Image Captioning as follows:
> | Metric                          | Flickr8k | THumB 1.0 | avg.  |
> |---------------------------------|----------|-----------|-------|
> | BLEU-1                          | 0.377    | 0.264     | 0.321 |
> | BLEU-4                          | 0.361    | 0.210     | 0.286 |
> | ROUGE                           | 0.375    | 0.237     | 0.306 |
> | METEOR                          | 0.481    | 0.281     | 0.381 |
> | CIDEr                           | 0.498    | 0.263     | 0.381 |
> | SPICE-F                         | 0.591    | 0.200     | 0.396 |
> | SPICE-PR                        | 0.603    | 0.170     | 0.387 |
> | SPICE-RE                        | 0.579    | 0.221     | 0.400 |
> | TIGEr                           | 0.542    | 0.153     | 0.348 |
> | CLIP-S                          | 0.514    | 0.292     | 0.403 |
> | RefCLIP-S                       | 0.576    | 0.210     | 0.393 |
> | MetaMetrics-Cap        |          |           |       |
> | GP (Full)                       | **0.638**| 0.298     | _0.468_|
> | GP (Top 5)                      | 0.560    | 0.246     | 0.403 |
> | XGBoost (Full)                  | 0.595    | **0.323** | 0.449 |
> | XGBoost (Top 5)                 | 0.575    | 0.311     | 0.443 |
> | XGBoost (Iterative-Best)        | 0.635  | 0.321   | **0.478** |
> | XGBoost (Iterative-Top 5)       | 0.634    | 0.301     | 0.467 |
>
> We hope this addresses your concern. Please feel free to reach out with any further questions; we are more than happy to assist.

---

> ### Comment · Reviewer_nKfe · 2024-11-24
>
> Thank you for addressing my questions. I appreciate the clarity of your in-domain experiments, where 30% of the dataset is used for training and the remaining 70% for evaluating the MetaMetrics.
>
> Since you are proposing a new evaluation metric, I’d like to confirm the intended protocol for using it. When using the MetaMetrics, it is always the ood setting, which is why I'm very concerned. Below is my understanding. Please correct me if this does not align with your intended approach. Some of my concerns and questions are based on this procedure:
> 1. Identification of the downstream task, such as Image Captioning.
> 2. Training MetaMetrics on datasets with available human ratings, evaluating on a held-out set, and selecting the MetaMetrics based on this in-domain test performance.
> 3. Applying the chosen MetaMetrics to a target ood dataset without human ratings to assess language model behavior against human preferences.
>
> Regarding your OOD experiments in the Robustness section, I have the following questions:
> 1. It’s encouraging to see experiments on unseen languages. However, I’m unclear about the experimental settings for robustness testing. Could you clarify:
>     + What are the language pairs (full dataset or 30%, and from which year) used for training MetaMetrics-MT?
>     + How was MetaMetrics-MT selected for evaluation on unseen languages (in-domain test performance on the rest 70%)?
>     + Additionally, what correlation metric is used in Table 17?
>     + Please also provide a reference for the WMT MQM 24 datasets.
> 2. Looking at Tables 16 and 17, it’s promising to see that MetaMetrics outperforms many conventional metrics individually. However, when comparing with all conventional metrics, the correlation metrics show marginal or no improvement. For cross-dataset evaluations, similar trends are observed.
> 3. Thank you for including Table 18. However, it appears to replicate Table 12 with less information. For example, the in-domain test performance, which is crucial for determining the appropriate version of MetaMetrics for OOD evaluations (based on the above mentioned evaluation protocol), is missing. As highlighted in Table 12 (MetaMetrics-Cap section, THumB1.0 column), the GP full model performs best in-domain testing. However, when applied to OOD datasets (MetaMetrics-Cap cross-dataset section), its performance (0.298) is very close to CLIP-S (0.292) and similar to METEOR (0.281).
>
> To summarize, I remain unconvinced that the MetaMetrics will consistently perform well in OOD settings. The beyesian optimization and boosting methods that are not inherently designed with ood generalization in mind. The ood generalization capability of metametrics may heavily depend on the task similarity (e.g., the ja-zh pair, which is dissimilar to any language pair in the training set, shows no improvement as reflected in Table 17).
>
> Thank you again for responding to my questions, and I am happy for further discussions.

---

> ### Author Response · Authors · 2024-11-24
> **Clarifying misconception (Part 1)**
>
> Thank you for your response. We'd like to help clarify any misunderstandings you may have.
>
> It seems there may have been some misunderstanding regarding certain aspects of our paper. Let us clarify the correct steps:
> 1. Identify the downstream task, such as image captioning.
> 2. Train MetaMetrics using datasets that include human ratings. Evaluate the performance on an **in-domain held-out set**, selecting the MetaMetrics based on the performance of this same **in-domain held-out set (not the test set). The test set is not used at this stage.**
> 3. Evaluate the selected MetaMetrics on **an unseen test dataset split that includes human ratings.** This is a **standard machine learning evaluation.** We use Kendall or Pearson correlation to assess language model behavior relative to human preferences. The test dataset split can include both in-domain unseen test data and out-of-domain unseen test data (e.g., cross-lingual and cross-dataset scenarios). **For the in-domain setting, use 70% of the dataset as the test split. For the out-of-domain setting, we use the whole dataset as test set.**
>
> Let's walk through a step-by-step example of testing in an out-of-domain cross-lingual setting for machine translation. This should help ensure we're on the same page.
> | #Size | Dataset A (zh-en) WMT20-22 | Dataset B (en-de) WMT20-22 | Dataset C (en-ru) WMT20-22 | Dataset D (he-en) WMT23 | Dataset E (en-es) WMT24 | Dataset F (ja-zh) WMT24 |
> | -------|------------|-------------|------------|-------------|------------|-------------|
> | train | A_train | B_train |C_train | Unavailable | Unavailable | Unavailable |
> | test | A_test | B_test |C_test | D_test |E_test | F_test |
>
> When evaluating on WMT23, we train our MetaMetrics using Datasets A_train, B_train, and C_train. During calibration, we fine-tune the MetaMetrics **only with the training data**. For testing, we apply our metric to the datasets D_test, E_test, and F_test, while ensuring that A_test, B_test, and C_test remain untouched. In this scenario, we do not have any training data for WMT23 or WMT24.
>
> Let's also walk through a step-by-step example for cross-dataset in case you need clarification.
> | #Size | Dataset A Flickr8k | Dataset B THumB1.0 |
> | -------|------------|-----------|
> | train | A_train | B_train |
> | test | A_test | B_test |
>
> When evaluating the Flickr8k on the **cross-dataset setting**, we train our MetaMetrics using Datasets B_train and test on A_train and A_test, and vice versa for THumB1.0.
>
> > What are the language pairs (full dataset or 30%, and from which year) used for training MetaMetrics-MT?
>
> As mentioned in the above example and F.7.1 Cross-lingual Setting Section in Appendix, here are the breakdown (just to make sure it is clear):
>
> | split | languages |
> | -------|------------|
> | train | WMT20-22: zh-en, en-de, en-ru |
> | test | WMT23: he-en, WMT24: en-es and ja-zh |
>
> > Additionally, what correlation metric is used in Table 17?
>
> Average Results of acc-t and Soft-pairwise accuracy (SPA). We reported the number shared with us by WMT24 organizers and added to the latest version of our paper.
>
> > Please also provide a reference for the WMT MQM 24 datasets.
>
> We added the citation of WMT24 shared task paper.  **To preserve anonymity, we will not specifically mention anything about the WMT24 paper.** We will add more information once this paper is accepted.

---

> ### Author Response · Authors · 2024-11-24
> **Clarifying misconception (Part 2)**
>
> > Looking at Tables 16 and 17, it’s promising to see that MetaMetrics outperforms many conventional metrics individually. However, when comparing with all conventional metrics, the correlation metrics show marginal or no improvement. For cross-dataset evaluations, similar trends are observed.
>
> It appears there may have been a misunderstanding regarding the interpretation of the tables. Let us clarify them for you. In Table 16, we demonstrate that MetaMetrics outperforms all individual metrics. When compared to XCOMET-Ensemble, MetaMetrics still shows **superior performance, although the margin is smaller**. It's important to note that, according to the WMT23 paper, **XCOMET-Ensemble utilizes much larger models** and they are larger than MetaMetrics. Additionally, we couldn't access their metric as the authors have not made it publicly available.
>
> In Table 17, the metrics we compared against are not the same as those used to build MetaMetrics. Many of these metrics are not currently accessible, as the shared task competition concluded only recently. We did not incorporate all metrics listed in Table 17 into MetaMetrics; they are included solely for comparison in relation to the WMT24 submission. It's important to note that MetricX-24 differs from MetricX-23, which we used during calibration. MetricX-24 is a more advanced version and is not publicly available. Therefore, we believe **it is not a fair comparison to claim there is no improvement**, as **we did not base MetaMetrics on these comparison metrics.** Furthermore, we employed the same MetaMetrics for both WMT23 and WMT24 without any additional calibration or tuning for adaptation.
>
> > However, it appears to replicate Table 12 with less information. For example, the in-domain test performance, which is crucial for determining the appropriate version of MetaMetrics for OOD evaluations (based on the above mentioned evaluation protocol), is missing. As highlighted in Table 12 (MetaMetrics-Cap section, THumB1.0 column), the GP full model performs best in-domain testing. However, when applied to OOD datasets (MetaMetrics-Cap cross-dataset section), its performance (0.298) is very close to CLIP-S (0.292) and similar to METEOR (0.281).
>
> Thank you for highlighting this issue. We want to emphasize that in **cross-dataset setting** even **our least performing results are still better than other existing metrics although it is not as far as our best metric.**
>
> > The beyesian optimization and boosting methods that are not inherently designed with ood generalization in mind.
>
> We have demonstrated that our **MetaMetrics** effectively **generalize across both cross-lingual and cross-dataset scenarios**. We chose Bayesian Optimization and Boosting Methods for their strengths in supervised learning, not explicitly for out-of-domain (OOD) generalization. These methods have shown potential for OOD performance if appropriately applied, as demonstrated in our work. Claiming that these method are not inherently designed with OOD generalization is **a strong statement without providing any references**. **While, in our experiments we showed that it is possible to generalize well on OOD.**

---

> ### Author Response · Authors · 2024-11-25
>
> Dear Reviewer nKfe,
>
> We hope this message finds you well. We kindly want to check if you had a chance to review our latest response and we have addressed all of your concerns, and if you have any further questions or comments we can address to help with your evaluation. Thanks again for your efforts and suggestions in improving our manuscript.
>
> Sincerely,
>
> The authors

---

> > ### Author Response · Authors · 2024-11-27
> > **Friendly Reminder: Upcoming Revision Deadline**
> >
> > As the revision deadline approaches, we kindly request your feedback on our rebuttal. We would greatly appreciate it if you could let us know whether we have successfully addressed your questions. Thank you in advance for your time and attention.
> >
> > Sincerely,
> >
> > The authors

---

> > > ### Comment · Reviewer_nKfe · 2024-11-27
> > >
> > > Thank you for providing additional explanations and clarifying the results, particularly in the out-of-domain (OOD) setting. Based on the context provided, I believe that the proposed method demonstrates sufficient evidence in the OOD setting.
> > >
> > > For the camera-ready version, I suggest incorporating some of the main results from the OOD setting into the main paper, as they are important in convincing readers to adopt your metric in their evaluation. Currently, Table 16 and 17 are hidden in the Appendix.
> > >
> > > Regarding the protocol, I think we align on the procedure. However, I have a question regarding Step 2, which involves using a held-out in-domain distribution set to choose the best-performed MetaMetrics. Since there are only two splits of the dataset (30% train and 70% test), which held-out set would you use if the 70% (test) set can not be used for in-domain evaluation?
> > >
> > > Based on our discussion, I have revised my score accordingly.
> > >
> > > Final mark: I would like to reiterate my point 3 - lack of evaluation in real setting - in the initial review.
> > > > For example, the authors could follow a similar evaluation procedure to DPO and PPO papers, using the proposed metric to evaluate various LLMs - before and after RLHF-tuned - and compare their win rates.
> > >
> > > The authors argue that their setting is in low-resource, but I think it is feasible if only conducting inference using pre-trained models. This real-world evaluation will significantly enhance the credibility of the proposed metric.
> > >
> > > I appreciate the efforts made by the authors during the discussion phase, including the experiments and explanations provided.

---

> > > > ### Author Response · Authors · 2024-11-27
> > > > **Thank you for your response and Further Clarification**
> > > >
> > > > Thank you for your insightful feedback. **As the revision period has concluded, we will incorporate the changes in the camera-ready version of the paper.** Below are our detailed responses:
> > > >
> > > > > For the camera-ready version, I suggest incorporating some of the main results from the OOD setting into the main paper, as they are important in convincing readers to adopt your metric in their evaluation. Currently, Table 16 and 17 are hidden in the Appendix.
> > > >
> > > > **Table 17 is actually a subset of Table 2.** **We will add clarifying text to the caption** to ensure readers understand this relationship. **Additionally, we are considering moving Table 16 to the main body of the document.**
> > > >
> > > > > Regarding the protocol, I think we align on the procedure. However, I have a question regarding Step 2, which involves using a held-out in-domain distribution set to choose the best-performed MetaMetrics. Since there are only two splits of the dataset (30% train and 70% test), which held-out set would you use if the 70% (test) set can not be used for in-domain evaluation?
> > > >
> > > > When we mention evaluating a held-out set, it means that this set is derived from the **training set**. We sample **20% data** from the **training set (from the original 30% split)** to create a **held-out (= 30% x 20% = 6% of all data)** set, and take the rest **80% data** split for training. Thus, **we never touch test set** during the **training and validation.**
> > > >
> > > > > evaluation in real setting - in the initial review.
> > > >
> > > > Regarding the evaluation in a real-world setting mentioned in your initial review, we decided to include additional experiments in the appendix of the **camera-ready version (since the revision period has ended)**. Specifically, we **will conduct experiments using reward models calibrated with MetaMetrics** and **assess the model's quality after aligning with human preferences**, as part of an ablation study. We believe **the experiment result will offer researchers further incentive to adopt our method when developing reward models.**

---

> ### Author Response · Authors · 2024-12-02
>
> Dear Reviewer nKfe,
>
> We hope this message finds you well. We kindly want to check if you had a chance to review our last rebuttal, and if you have any final comments after our last response. Thank you again for your review.
>
> Sincerely,
>
> The authors

---

### Official Review · Reviewer_HGah · 2024-11-04

**Soundness:** 3
**Presentation:** 3
**Contribution:** 3
**Rating:** 8
**Confidence:** 4

**Summary:**

This paper introduces Meta-Metrics, a calibrated metric for evaluating generation tasks like summarization, QA, Machine Translation and Image Captioning,etc. It raises the issue of metrics excelling at one dimension and falling at others while evaluating on multiple dimensions (like coherence, fluency, relevance,etc in summeval), hence proposing a calibrated metric based on these several metrics, aiming to achieve a higher alignment with human preferences in the evaluation scores. The authors propose using methods like Bayesian Optimization, Boosting to effectively choose/filter the various metrics and give a combined metric to have higher correlation with human scores.

**Strengths:**

1. The paper presents innovative mathematical approaches of using BO modeled as Gaussian Process, and XG Boosting on existing metrics to create a new metric.
2. The paper has tested their method extensively on several datasets across multiple domains.

**Weaknesses:**

1. Although the metametrics have shown they have higher alignment with human preferences, the optimization methods (BO and Boosting) are to be considered black boxes and offer very less explainability.  Understanding how these methods find the combination of metrics and interactions between different metrics within the model is not possible. Hence explanation on why they correlate more with human scores is not there.

**Questions:**

1. It is very good that the authors have shown their method on multiple datasets, but some more experiments/analysis on the proposed methods would strengthen the paper more. One such thing can be, the authors used Kendall-Tau as the objective calibration function. How were the results when spearman correlation is used as a calibration function?

---

> ### Author Response · Authors · 2024-11-20
> **Ran Additional Experiments for Ablation Study and Clarification on Interpretability**
>
> Thank you for useful review. We have addressed your questions and concerns as follows:
>
> > It is very good that the authors have shown their method on multiple datasets, but some more experiments/analysis on the proposed methods would strengthen the paper more. One such thing can be, the authors used Kendall-Tau as the objective calibration function. How were the results when spearman correlation is used as a calibration function?
>
> Following your suggestion, we have **added an ablation study** in Appendix F.4 using Spearman correlation as the calibration function on XGBoost for the Summarization task. The results show **minimal performance differences** between the two evaluation functions, which is expected given the similarity of evaluation measurement between Spearman and Kendall correlations.
>
> ```markdown
> | Metric             |                SummEval           |     Benchmark LLM     |    Combined   | Avg.  |
> |                    | Coh.  | Cons. | Fluency | Rel.    | Faith.| Coh.  | Cons. | Coh.  | Cons. |       |
> |--------------------|-------|-------|---------|---------|-------|-------|-------|-------|-------|-------|
> | XGBoost (Kendall)  | 0.190 | 0.180 |  0.164  |  0.255  | 0.406 | 0.411 | 0.397 | 0.403 | 0.367 | 0.308 |
> | XGBoost (Spearman) | 0.185 | 0.174 |  0.165  |  0.261  | 0.406 | 0.411 | 0.397 | 0.402 | 0.369 | 0.308 |
> ```
>
> > Although the metametrics have shown they have higher alignment with human preferences, the optimization methods (BO and Boosting) are to be considered black boxes and offer very less explainability. Understanding how these methods find the combination of metrics and interactions between different metrics within the model is not possible. Hence explanation on why they correlate more with human scores is not there.
>
> Optimization methods like BO retain **a level of interpretability**, particularly in terms of **weight allocation**, and are **well-motivated and transparent** in their operation. Although boosting is comparatively less interpretable than BO, it still allows us to assess metric importance through **feature importance**. Thus, these enable us to **understand the contribution of each metric** to the overall performance metric.

---

> > ### Comment · Reviewer_HGah · 2024-11-22
> >
> > Thank you for the rebuttal.
> >
> > > Following your suggestion, we have added an ablation study in Appendix F.4 using Spearman correlation as the calibration function on XGBoost for the Summarization task. The results show minimal performance differences between the two evaluation functions, which is expected given the similarity of evaluation measurement between Spearman and Kendall correlations.
> >
> > Thank you for performing this experiment and clearing the doubt. The results looks good to me.
> >
> > > Although boosting is comparatively less interpretable than BO, it still allows us to assess metric importance through feature importance. Thus, these enable us to understand the contribution of each metric to the overall performance metric.
> >
> > Would you mind elaborating more on this as this is not clear yet ?

---

> > > ### Author Response · Authors · 2024-11-22
> > >
> > > Thank you for your response. We’d be happy to address your follow-up question.
> > >
> > > We currently use XGBoost as our boosting method, which provides a way to assess the importance of each feature in the model. Feature importance measures the contribution of each feature to the model's predictive power. In XGBoost, feature importance is determined by evaluating how each feature influences the model's decisions during training. This process is rooted in the construction of decision trees, where each split is based on a single feature. As the trees are built, XGBoost tracks the improvement in model performance resulting from each split, typically measured using metrics like information gain, which quantifies how much uncertainty or error is reduced by the split.
> > >
> > > By aggregating these improvements across all trees, XGBoost assigns an importance score to each feature, highlighting the most impactful ones. However, unlike methods like Bayesian Optimization (BO), XGBoost does not provide a simple closed-form expression, such as $\sum_{i=1}^{N} \hat{y_{i}}w_{i}$, to explicitly show how individual features contribute to the overall model performance. While feature importance in XGBoost offers valuable insights into the relative importance of features, it is numerically less interpretable compared to the more transparent, explicit relationships provided by methods like BO.

---

> > > > ### Comment · Reviewer_HGah · 2024-11-24
> > > > **Additonal Questions**
> > > >
> > > > Thanks a lot for your explanation.
> > > >
> > > > Upon revisiting the paper, I have few other questions.
> > > >
> > > > 1) For Table-1, I would like to understand why there are no baselines in reference-free metrics like G-Eval (https://arxiv.org/abs/2303.16634) or any of the latest works. They are straight-forward to apply and would show correlation of the range (0.5 +). Kindly justify if the work cannot be a baseline or show its results as a baseline.
> > > > 2) Line 220-221 "Our METAMETRICS-SUM correlates the best with human ratings across different datasets and aspects of evaluation." usually correlation is checked with the average rating of the human annotators why did you check only with the best human annotator rating ? can you just do the meta evaluation (i.e finding the correlation on your results again) with the average rating instead of best ?

---

> > > > > ### Author Response · Authors · 2024-11-25
> > > > > **Added G-Eval + UniEval and clarification**
> > > > >
> > > > > Thank you for your positive, constructive suggestions and keen observations. We greatly appreciate your questions and the support you've provided in improving our paper. We will address your points as follows:
> > > > >
> > > > > > For Table-1, I would like to understand why there are no baselines in reference-free metrics like G-Eval (https://arxiv.org/abs/2303.16634) or any of the latest works. They are straight-forward to apply and would show correlation of the range (0.5 +). Kindly justify if the work cannot be a baseline or show its results as a baseline.
> > > > >
> > > > > Thank you for mentioning G-Eval, a GPT-4-based metric. We evaluated our SummEval benchmark **with a distinct test set split**, different from the original SummEval setup. As a result, the numbers we report differ from those in their papers. Specifically, we employed the same setup as previously outlined in the paper, allocating 30% of the dataset for training and calibrating MetaMetrics and 70% for testing. Consequently, we **reran the evaluation using the scores provided in the authors' GitHub repository**.
> > > > >
> > > > > To make the benchmark more comprehensive, we also included **UniEval** and selected both **G-Eval and UniEval** as metrics for calibrating MetaMetrics, and we also reported **individual results**. This resulted in **significantly improved MetaMetrics performance**, and we have updated Table 1 accordingly. Our results are summarized as follows:
> > > > >
> > > > > | Metric             | Coh. | Cons. | Fluency | Rel.  | avg   |
> > > > > |--------------------|-------|-------|-------|-------|-------|
> > > > > | BLEU               | 0.110 | 0.126 | 0.113 | 0.170 | 0.130 |
> > > > > | chrF               | 0.143 | 0.094 | 0.071 | 0.198 | 0.127 |
> > > > > | METEOR             | 0.077 | 0.102 | 0.072 | 0.162 | 0.103 |
> > > > > | ROUGE-WE1          | 0.115 | 0.088 | 0.081 | 0.169 | 0.113 |
> > > > > | BARTScore (Mean)   | 0.086 | 0.074 | 0.040 | 0.143 | 0.086 |
> > > > > | BLEURT (Max)       | 0.185 | 0.070 | 0.114 | 0.189 | 0.140 |
> > > > > | BERTScore (f1)     | 0.105 | 0.100 | 0.120 | 0.181 | 0.127 |
> > > > > | UniEval            | 0.413 | 0.353 | 0.359 | 0.324 | 0.362 |
> > > > > | G-Eval             | 0.429 | 0.413 | 0.409 | 0.437 | 0.422 |
> > > > > | **MetaMetrics** |
> > > > > | GP                 | 0.455 | 0.411 | 0.409 | **0.449** | 0.431 |
> > > > > | GP (UniEval + G-Eval)         | 0.462 | 0.428 | 0.409 | **0.449** | 0.437 |
> > > > > | XGBoost            | **0.475** | 0.376 | 0.407 | 0.439 | 0.424 |
> > > > > | XGBoost (UniEval + G-Eval)    | **0.475** | **0.430** | **0.430** | 0.446 | **0.445** |
> > > > >
> > > > > Overall, our results show that **MetaMetrics outperforms all individual metrics.** Please note that GPT-4-based metrics (e.g., G-Eval) may have concerns regarding **data contamination and leakage**, given that the datasets and their labels (human scores) are publicly available online and have been used to finetune LLMs. Including such a metric as a baseline could lead to an unfair comparison.
> > > > >
> > > > > > Line 220-221 "Our METAMETRICS-SUM correlates the best with human ratings across different datasets and aspects of evaluation." usually correlation is checked with the average rating of the human annotators why did you check only with the best human annotator rating ? can you just do the meta evaluation (i.e finding the correlation on your results again) with the average rating instead of best ?
> > > > >
> > > > > In our experiment, we used the **average rating of the human annotator scores** rather than the rating from the best human annotator. We have updated the paper to improve the clarity.

---

> > > > > > ### Comment · Reviewer_HGah · 2024-11-25
> > > > > >
> > > > > > Thank you very much for your additional experiments but now this raises more concerns so kindly change the presentation.
> > > > > >
> > > > > > 1. Please don't include Reference free metrics like G-Eval for calibrating metametrics, as this will definitely increase the overall correlation because Reference free metrics like G-Eval tend to have better correlations than reference based metrics like Bleu or meteor etc and it is not fair to compare them. I only wanted this baselines to be included so that we can see the current difference in performance between reference free zero shot methods like G-Eval (which uses LLMs like GPT-4 as backbone) vs reference based (correct me if i am wrong, but as far as I understand metametrics is a reference based evaluation).
> > > > > > 2. Also when comparing with G-Eval kindly mention explicitly all the hyperparams used, exact model name.
> > > > > > 3. G-Eval explains in their paper that Spearman Correlation maybe a better metric to compare scores, so it will be interesting to show both spearman and kendaltau in your table-1.
> > > > > >
> > > > > >
> > > > > > I believe that metametrics can be a very interesting approach if you can back these things up because it a non LLM based method hence the cost and speed of evaluation is much better than LLM based evaluation. The only problem with this methodology is that this is a training and reference based method while other LLM methods are only zero shot reference free methods. This is still valuable because LLM based methods may not be scalable when we want to evaluate thousands of samples.
> > > > > >
> > > > > > > Overall, our results show that MetaMetrics outperforms all individual metrics. Please note that GPT-4-based metrics (e.g., G-Eval) may have concerns regarding data contamination and leakage, given that the datasets and their labels (human scores) are publicly available online and have been used to finetune LLMs. Including such a metric as a baseline could lead to an unfair comparison.
> > > > > >
> > > > > > The comparision is definitely unfair because of the underlying methodological differences and not because of data contamination (it will be great if you can cite something to say why you think data contamination happened). G-Eval is method that can be potentially applied with different LLMs if that is a challenge.
> > > > > >
> > > > > > I would also like to see the overall time taken for evaluating a dataset (you can choose summeval) and then I want you to compare it with other methods (incl G-Eval although the original G-Eval code does sequentially without multiple workers or parallizability while calling LLM api, it is fine to just understand the overall time taken for a evaluation) Also mention the time it takes for training and what resources are needed for it.
> > > > > >
> > > > > > If you can change the presentation and show these all requested results then I will be more than happy to increase my score.
> > > > > >
> > > > > > Overall i believe this is a great work when presented in the right sector i.e training and reference based method which is cost effective and potentially faster than LLM based methods.
> > > > > >
> > > > > > Thanks

---

> > > > > > > ### Author Response · Authors · 2024-11-26
> > > > > > > **Updated Results Table Presentation, Included Spearman Results, and Detailed Training and Inference Time/Cost**
> > > > > > >
> > > > > > > Thank you for your constructive and positive feedback. We appreciate the time and effort you have invested in collaborating with us to enhance the quality of our paper. **We have addressed all of your requests and concerns**, and have **updated our paper's presentation and added spearman results, and detailed training and inference time/cost** accordingly. Below, we provide detailed responses to your questions and concerns:
> > > > > > >
> > > > > > > > correct me if i am wrong, but as far as I understand metametrics is a reference based evaluation
> > > > > > >
> > > > > > > We would like to emphasize that **MetaMetrics supports both reference-based and reference-free metrics, including the combination of both.** This flexibility allows MetaMetrics to harness the strengths of individual metrics, combining them into a metric that is more closely aligned with human preferences.
> > > > > > >
> > > > > > > > Change of presentation and added spearman correlation
> > > > > > >
> > > > > > > **We have updated Table 1 to include the baseline metrics and Spearman correlation results.** Additionally, we report **MetaMetrics under two settings**: **with** and **without LLM-based metrics.** This allows for a comprehensive comparison between different metrics. We also reran experiments and summarized the results as follows:
> > > > > > >
> > > > > > > | **Metric**       | Kendall |     |         |   |     |  Spearman |    |        |    |     |
> > > > > > > |-------|-----|---|----|---|-----|------------|-------|------|---|--------|
> > > > > > > |        | **Coh.** | **Cons.** | **Fluency** | **Rel.** | **Avg.** | **Coh.** | **Cons.** | **Fluency** | **Rel.** | **Avg.** |
> > > > > > > | **LLM-based Metrics**  |
> > > > > > > | BARTScore (Mean)      | 0.086  | 0.074  | 0.040  | 0.143  | 0.086  | 0.123  | 0.094  | 0.052  | 0.202  | 0.118  |
> > > > > > > | UniEval    | 0.413  | 0.353  | 0.359  | 0.324  | 0.362  | 0.577  | 0.439  | 0.458  | 0.446  | 0.480  |
> > > > > > > | G-Eval (GPT4)         | 0.429  | 0.413  | _0.409_ | 0.437  | 0.422  | 0.565  | 0.510  | 0.470  | 0.581  | 0.531  |
> > > > > > > | **MetaMetrics-Sum** |                                                                                                         |
> > > > > > > | GP (all without LLM)      | 0.172  | 0.140  | 0.130  | 0.252  | 0.174  | 0.244  | 0.179  | 0.167  | 0.354  | 0.236  |
> > > > > > > | XGBoost               | 0.192  | 0.186  | 0.186  | 0.276  | 0.210  | 0.274  | 0.236  | 0.239  | 0.386  | 0.284  |
> > > > > > > | **MetaMetrics-Sum w/ LLM-based Metrics**  |
> > > > > > > | GP  (all with LLM)    | 0.454  | 0.419  | _0.409_ | **0.449** | 0.433  | 0.609  | _0.519_ | 0.470  | **0.601** | 0.550  |
> > > > > > > | GP (Top 2)          | _0.461_ | _0.428_ | _0.409_ | **0.449** | _0.437_ | 0.628  | **0.528** | 0.470  | **0.601** | _0.557_ |
> > > > > > > | XGBoost   (all with LLM)           | **0.476** | 0.367  | 0.404  | _0.447_ | 0.424  | **0.642** | 0.460  | **0.512** | _0.600_ | 0.553  |
> > > > > > > | XGBoost (Top 2)   | **0.476** | **0.436** | **0.430** | 0.445  | **0.447** | _0.636_ | 0.508  | _0.511_ | 0.594  | **0.562** |
> > > > > > >
> > > > > > > > Time and Cost
> > > > > > >
> > > > > > > **We have added a new table in Appendix Table 20, Section F.8**, which provides a **detailed breakdown of the time costs for calibration and inference in sequence.** The breakdown includes inference time and calibration time. We ran the benchmarking on the same machine
> > > > > > >
> > > > > > > **The calibration time for MetaMetrics is very quick**, taking **up to 647.08 seconds (about 11 minutes)**. In contrast, using the OpenAI API for G-Eval (GPT4) required considerably more time when running the inference on test set, which is **7,173.50 seconds** (approximately 11x longer than the calibration), and **it showed that the bottleneck comes from the inference rather the calibration** and **G-Eval (GPT4) cost is around US$82 for SummEval.** MetaMetrics without LLM-based metrics is much more efficient in terms of the inference time compared to MetaMetrics with LLM-based metrics.
> > > > > > >
> > > > > > > We ran the G-Eval code using the original repository without any modifications. Also, **the reported evaluation time for the training and test splits in MetaMetrics uses a sequential evaluation of metrics to ensure fairness, as this is consistent with the sequential process of G-Eval**. This sequential evaluation can be optimized by running metrics in parallel **(which are not reported in Table 20)**, as their evaluations are completely independent of each other. We highlight some of the results here:
> > > > > > >
> > > > > > > | **Metric**     | **Inference (Train Split) (in sec)** | **Inference (Test Split) (in sec)** |  **Calibration (in sec)** |
> > > > > > > |-----|---|--------|-----|
> > > > > > > | G-Eval (GPT4)     | 3,081.46 | 7,173.50 | N/A     |
> > > > > > > | **MetaMetrics-Sum**        |       |    |    |
> > > > > > > | GP (all without LLM-based metric)    | 792.25   | 1,873.95  | 402.51  |
> > > > > > > | XGBoost (all without LLM-based metric)    | 792.25    | 1,873.95     | 276.52  |
> > > > > > > | **MetaMetrics-Sum w/ LLM-based Metrics** |      |        |         |
> > > > > > > | GP (all with LLM-based metric)            | 3,976.47 | 9,292.83| 647.08 |
> > > > > > > | XGBoost (all with LLM-based metric)       | 3,976.47| 9,292.83 | 337.88 |
> > > > > > >
> > > > > > > We hope that our response has addressed all of your questions. Thank you once again for your insightful comments and suggestions.

---

> > > > > > > > ### Comment · Reviewer_HGah · 2024-11-26
> > > > > > > >
> > > > > > > > Thank you very much for all the latest changes, now the structure looks much better and i am more confident on my evaluation.
> > > > > > > >
> > > > > > > > Final question for clarity:
> > > > > > > > 1. Did you utilize the same prompts given in the G-Eval repo ? and used the same hyperparams including n=20 ?
> > > > > > > >
> > > > > > > > And for all the experiments concerning G-Eval, you don't really have to utilize GPT-4, you can point to any latest openai LLM like GPT-4o-mini to minimize the cost.
> > > > > > > >
> > > > > > > > I am increasing the score by 2 points and increasing my confidence too.
> > > > > > > >
> > > > > > > > **Final conclusion**: It appears that MetaMetrics is really good because it can even utilize LLM based evaluations and scale to both reference-free and reference-based evaluations and the inference speed is justified given the improvement in correaltions.   One interesting advantage of this method is that it can always improve any new metric's score by calibrating. The only drawback is that it involves training or calibration (which is fair enough given the improvement in the correaltions) But can effect in practical application where we don't have data to calibrate the MetaMetrics as it involves understanding of the given response to score it. MetaMetrics falls in the intersection of reference-free and reference-based training based method and can be potentially very helpful or impactful for evaluation domain.
> > > > > > > >
> > > > > > > > I appreciate authors for their excellent clarity and efforts.

---

> > > > > > > > > ### Author Response · Authors · 2024-11-26
> > > > > > > > >
> > > > > > > > > Thank you for your positive feedback. We truly appreciate it. In response to your final question, yes, we use the same prompts and hyperparameters, including n=20.

---

### Official Review · Reviewer_y9ei · 2024-11-05

**Soundness:** 2
**Presentation:** 3
**Contribution:** 2
**Rating:** 6
**Confidence:** 4

**Summary:**

This paper introduces meta-metrics, which optimizes the combination of existing metrics to enhance their alignment with human preference. The paper investigated mainly two method GP and XGBoost to try to learn a best weight vector for a set of base metrics to combine. For different datasets, they select different base metrics and train a new weight vector for combinations. Results show that meta-metrics can outperform existing baseline metrics in summarization, translation, question answering, image captioning and reward modeling scenarios.

**Strengths:**

1. This paper proposes a new paradigm to ensemble existing metrics for evaluation of different tasks, learning a weight vector through GP or XGBoost and get the weighted combination of base metrics. This is a novel approach and gets good performance.
2. The paper is well-motivated and clearly written, with nice figures and tables illustrating the concepts.
3. Extensive experiments across summarization, machine translation, QA, reward modeling, which covered a wide areas. The authors also provide analysis of the weights distribution across the base metrics. Providing many insights.

**Weaknesses:**

1. The performance gain is relatively small compared to base metrics, especially for machine translation and reward modeling tasks, it's unclear whether these marginal performance gains are worth the training cost.
2. The proposed approach requires training a new combination weight vector once the base metrics have changed, thus limiting the usability of the approach.

**Questions:**

1. Please provide the training details. From my understanding, when training a combined metametric for reward modeling, you need to run inference on all of the training data using each of the base metrics first, and then train your weight vector to optimize for the human preference. How much GPU hours have you spent on the inference, and the training? How many GPUs did you use for the experiments?
2. Could you provide a list of base metrics used for each meta-metric variant in one place? I only found the base metrics for reward modeling  in line 474 and it's unclear what's the base metrics used for other benchmarks. if they are also baselines, it's better to mark them in the results table.

---

> ### Author Response · Authors · 2024-11-20
> **Added More Details for Clarity, Comprehensive List for Base Metrics, and Clarification on Training and Inference**
>
> Thank you for your insightful review. We have addressed your questions and concerns as follows:
>
> > The proposed approach requires training a new combination weight vector once the base metrics have changed, thus limiting the usability of the approach.
>
> **In terms of efficiency**, the calibration process is lightweight, requiring only CPU resources, and scores for each base metric from previous runs can be reused even if there are changes to the set of base metrics used for MetaMetrics, eliminating the need for additional GPU resources.
>
> In addition, **regarding metric robustness**, our approach demonstrates adaptability to unseen domains and languages, as evidenced by our results in image captioning and machine translation tasks. This allows us to apply the same weight combination to similar tasks across different domains or languages.
>
> > Please provide the training details. From my understanding, when training a combined metametric for reward modeling, you need to run inference on all of the training data using each of the base metrics first, and then train your weight vector to optimize for the human preference. How much GPU hours have you spent on the inference, and the training? How many GPUs did you use for the experiments?
>
> To perform the entire inference process to obtain a score using GPUs, we generally require about a day when employing the A100 40GB for reward modeling, as noted in the Limitations section. We allocate one metric per GPU for LLM-based metrics. In contrast, non-LLM-based metrics are executed on CPUs. Once all the scores are acquired, the calibration process takes approximately 30 minutes to an hour for GP and about 10 minutes for XGBoost on a 16-core CPU.
>
> > Could you provide a list of base metrics used for each meta-metric variant in one place? I only found the base metrics for reward modeling in line 474 and it's unclear what's the base metrics used for other benchmarks.
>
> For clarity, we have added a comprehensive list of the base metrics in Appendix Section G. This section is also accompanied by heat maps that illustrate the weights of each feature or metric for both GP and XGBoost.

---

> > ### Author Response · Authors · 2024-11-24
> >
> > Dear Reviewer y9ei,
> >
> > We hope this message finds you well. We kindly want to check if you had a chance to review our rebuttal, and if you have any further questions or comments we can address to help with your evaluation. Thanks again for your efforts and suggestions in improving our manuscript.
> >
> > Sincerely,
> >
> > The authors

---

> > ### Comment · Reviewer_y9ei · 2024-11-24
> > **Response to author's rebuttal**
> >
> > Thanks for the author's rebuttal. I acknowledge the additional contents provided in the appendix, which I think is a great supplement for readers to understand the paper. Generally, I think the current rating is fair and decided not to change the rating.

---

### Official Review · Reviewer_H1Lp · 2024-11-10

**Soundness:** 3
**Presentation:** 3
**Contribution:** 3
**Rating:** 6
**Confidence:** 3

**Summary:**

This paper introduces MetaMetrics, a framework designed to calibrate evaluation metrics to better align with human preferences. By leveraging existing metrics that capture different aspects of human preferences, MetaMetrics employs learning-based methods to optimize these metrics for various generative tasks. To demonstrate the effectiveness of MetaMetrics in aligning with human preferences, the authors compute correlations between the calibrated metrics and quantified human ratings. Experiments are conducted across diverse tasks, including Text Summarization, Machine Translation, Question Answering, Image Captioning, and Reward Model Scoring.

**Strengths:**

- The paper addresses a critical problem in the field of text generation: the lack of automatic metrics that accurately reflect human preferences. Current approaches often rely heavily on human annotations to evaluate performance. By proposing a method to calibrate existing metrics to align with human preferences, this work aims to reduce the reliance on labor-intensive human evaluation processes.

- The methodology is simple yet effective. The proposed approach captures the multifaceted nature of human preferences while incurring minimal additional cost, enabling efficient calibration of existing metrics.

- Extensive experiments validate the proposed method. The authors conduct experiments across a variety of tasks, including Text Summarization, Machine Translation, Question Answering, Image Captioning, and Reward Model Scoring. This demonstrates the task-agnostic advantages of the method and its broad applicability.

**Weaknesses:**

- Empirical and ad hoc nature of MetaMetrics: The calibration process for MetaMetrics relies on empirical and ad hoc weighting of existing metrics for each dataset and human preference. These weights are highly dependent on the specific dataset and task, which raises concerns about their generalizability to a broader range of datasets or tasks.

- Lack of ensembled baselines: While the paper compares MetaMetrics against individual metrics, it does not include ensembled baselines, such as averaged or weighted-average scores of individual metrics. Including these as baselines would provide a fairer comparison and strengthen the evaluation.

- Absence of human studies in experiments: Although MetaMetrics aims to align with human preferences in evaluating model performance, the study does not involve an actual model being evaluated and analyzed through human studies. To comprehensively validate MetaMetrics, experiments involving direct human evaluation of model outputs are necessary.

- Unclear human preference in certain tasks: In some evaluation tasks, the connection between the ratings and human preferences is not adequately explained. For instance, it is unclear how ratings like sys Pearson, seg Pearson, and acc-t in Machine Translation tasks, or NQ, TQ, NarrativeQA, and SemEval in Question Answering tasks, align with human preferences. More detailed explanations of these ratings and their relationship to human preferences are needed, especially for readers unfamiliar with these tasks.

- High computational cost for reward model ensembling: The use of multiple metrics in MetaMetrics can significantly increase evaluation costs, particularly in reward model scoring. For example, in online RLHF training, simultaneously running multiple reward models (e.g., 7 models in Table 4) could make the process computationally infeasible.

**Questions:**

See weaknesses.

---

> ### Author Response · Authors · 2024-11-20
> **Added Baselines and Clarifications on Metric Generalizability, Effectiveness, and Efficiency**
>
> Thank you for your constructive suggestion. We have addressed your questions and concerns as follows:
>
> > Empirical and ad hoc nature of MetaMetrics: The calibration process for MetaMetrics relies on empirical and ad hoc weighting of existing metrics for each dataset and human preference. These weights are highly dependent on the specific dataset and task, which raises concerns about their generalizability to a broader range of datasets or tasks.
>
> To enhance the **generalizability** of our method, we have applied MetaMetrics to a diverse range of tasks and modalities, including abstract summarization, machine translation, question answering, reward modeling, and image captioning. We also test the robustness of our method in **cross-domain and cross-lingual contexts**. Specifically, for image captioning, we calibrate our metric in one domain and evaluate it in an unseen domain, as illustrated in Figure 2. This demonstrates that MetaMetrics remains robust in unfamiliar domains. Additionally, we carry out experiments on cross-lingual machine translation tasks by evaluating our model with an **unseen language pair (he-en)**. In this paper, we limit our exploration for a single model per task and the development of a single model applicable to all tasks is left for future research.
>
> > Lack of ensembled baselines: While the paper compares MetaMetrics against individual metrics, it does not include ensembled baselines, such as averaged or weighted-average scores of individual metrics. Including these as baselines would provide a fairer comparison and strengthen the evaluation.
>
> Thank you for your suggestion. We have included baselines for both **uniform averaging** and **weighted averaging** across all tasks to ensure a fairer comparison. Our method outperforms these baselines across all tasks.
>
> > Absence of human studies in experiments: Although MetaMetrics aims to align with human preferences in evaluating model performance, the study does not involve an actual model being evaluated and analyzed through human studies. To comprehensively validate MetaMetrics, experiments involving direct human evaluation of model outputs are necessary.
>
> All of our test sets are **all human scores** and we use kendall correlation to compute the correlation between our scores and humans. We report all results against humans.
>
> > Unclear human preference in certain tasks: In some evaluation tasks, the connection between the ratings and human preferences is not adequately explained. For instance, it is unclear how ratings like sys Pearson, seg Pearson, and acc-t in Machine Translation tasks, or NQ, TQ, NarrativeQA, and SemEval in Question Answering tasks, align with human preferences. More detailed explanations of these ratings and their relationship to human preferences are needed, especially for readers unfamiliar with these tasks.
>
> We have provided a detailed description for each rating, including **kendall, sys Pearson, seg Pearson, and acc-t** in Appendix Section H to enhance clarity on how ratings are aligned with human preferences.
>
> > High computational cost for reward model ensembling: The use of multiple metrics in MetaMetrics can significantly increase evaluation costs, particularly in reward model scoring. For example, in online RLHF training, simultaneously running multiple reward models (e.g., 7 models in Table 4) could make the process computationally infeasible.
>
> MetaMetrics is designed to be **embarrassingly parallel**, enabling each metric to be executed independently without requiring inter-model communication, which results in exceptional efficiency. Thus, making it much efficient to run inference if there are multiple models used in MetaMetrics. To have a fair comparison, we take models with similar performance, such as the 70B ones like Skywork-Critic-Llama-3.1-70B, TextEval-Llama3.1-70B, and SFR-LLaMa-3.1-70B-Judge-r. Our MetaMetrics (GP) reward model, with only 41B parameters (composed of multiple models where **the largest model contains 8B parameters**), **outperformed these 70B models**. Additionally, despite its size, our MetaMetrics reward model allows for parallel execution of individual models with a computational cost equivalent to 8B parameters, enabling us to use considerably less compute compared to the 70B models, which demand much more substantial resources. Additionally, **our metric can still run in a GPU with smaller computing requirement such as A100 40GB**, while the 70B model requires much larger compute with 80GB GPU with half precision.

---

> > ### Author Response · Authors · 2024-11-24
> >
> > Dear Reviewer H1Lp,
> >
> > We hope this message finds you well. We kindly want to check if you had a chance to review our rebuttal, and if you have any further questions or comments we can address to help with your evaluation. Thanks again for your efforts and suggestions in improving our manuscript.
> >
> > Sincerely,
> >
> > The authors

---

> > > ### Comment · Reviewer_H1Lp · 2024-11-27
> > >
> > > Thank you to the authors for their detailed clarifications. Most of my major concerns have been addressed. However, I remain concerned about the lack of real model evaluations. I strongly encourage the authors to include at least one ablation study, such as evaluations of the LLMs before and after alignment, in the camera-ready version. Implementing this should not require significant additional effort or resources. Based on the revisions, I have raised my score from 5 to 6.

---

> ### Author Response · Authors · 2024-11-27
> **Thank you for your feedback and raising the score. We will add the ablation study in the camera-ready version.**
>
> Thank you for your insightful comments and for helping to enhance our score. We find your suggestions feasible and are pleased to incorporate the proposed ablation study in the camera-ready version. We also believe this idea will provide researchers with additional motivation to adopt our method when developing reward models.

---

### Official Review · Reviewer_4VeV · 2024-11-11

**Soundness:** 3
**Presentation:** 3
**Contribution:** 3
**Rating:** 6
**Confidence:** 3

**Summary:**

This paper proposed a method to align the automatic evaluation metrics with human preference evaluation, called MetaMetrics. Specifically, it's optimized to learn a set of weights that can combine automatic metrics to mimic human preference. Two methods are discussed and experimented for the optimization. The experiments cover both text and multimodal evaluation, demonstrating a satisfactory performance of the proposed method.

**Strengths:**

1. The research problem this paper studied is very interesting and useful. Automatic evaluation metrics can cover a wide range of benchmarks. However, they may not align with human evaluation. Given human evaluation is time-consuming and expensive, and also current model is required to be more generalizable to various tasks, aligning the automatic metrics with human evaluation preference becomes an urgent and important research topic.

2. This paper illustrates the motivation, problem definition, and methods to optimize the proposed MetaMetrics in a clear and easy-to-follow way.

3. The experiments are conducted in a wide range of downstream tasks, from text to multimodal, as well as training a reward model.

**Weaknesses:**

1. This paper assumes that human preference can be approximated by a weighted combination of metric scores. But it seems to be too simple, and might not always hold. For example, human's evaluation of helpfulness might require both the model to perform well in reasoning and knowledge-retrieving, which we assume are separately measured by two benchmarks. Then a weighted summation of these two benchmarks might not be suitable, instead, a multiplication might be more proper.

**Questions:**

See weaknesses.

---

> ### Author Response · Authors · 2024-11-20
> **Highlighted robustness evaluation and provided multiplicative weighting scores**
>
> Thank you for your constructive suggestion. We have addressed your questions and concerns as follows:
>
> > This paper assumes that human preference can be approximated by a weighted combination of metric scores. But it seems to be too simple, and might not always hold. For example, human's evaluation of helpfulness might require both the model to perform well in reasoning and knowledge-retrieving, which we assume are separately measured by two benchmarks.
>
> To ensure the generalizability of our method, we apply MetaMetrics to a **variety of tasks and modalities**, including abstract summarization, machine translation, question answering, reward modeling, and image captioning. We also explore the robustness of our method in **cross-domain** and **cross-lingual** scenarios. For image captioning, we calibrate our metric in one domain and test it in an unseen domain, as shown in Figure 2, demonstrating that MetaMetrics maintains robustness in **unfamiliar domains**. Additionally, we conduct experiments on **cross-lingual machine translation tasks** by evaluating our model with an **unseen language pair (he-en)**.
>
> > Then a weighted summation of these two benchmarks might not be suitable, instead, a multiplication might be more proper.
>
> Following your suggestion, in Appendix Section F.6, we have included a **new ablation study** comparing **multiplicative and linear weighting**. Our findings indicate that linear weighting generally perform better or on par to the multiplicative approach. We conjecture that the multiplicative method introduces a higher-dimensional space and GP may not perform well on higher-dimensional space [1,2,3,4,5], especially on Summarization and Reward Model.
>
> ```
> | Method                  | Summarization | MT    | MT (QE) | Reward Model |
> |-------------------------|---------------|-------|---------|--------------|
> | GP (Linear)             | 0.262         | 0.819 | 0.801   | 93.5         |
> | GP (Multiplicative)     | 0.238         | 0.822 | 0.798   | 91.2         |
> | GP (Combined)           | 0.234         | 0.818 | 0.803   | 92.1         |
> ```
>
> **References:**
> - [1] Frazier, P. I. (2018). A tutorial on Bayesian optimization. arXiv preprint arXiv:1807.02811.
> - [2] Binois, M., & Wycoff, N. (2022). A survey on high-dimensional Gaussian process modeling with application to Bayesian optimization. ACM Transactions on Evolutionary Learning and Optimization, 2(2), 1-26.
> - [3] Li, C. L., Kandasamy, K., Póczos, B., & Schneider, J. (2016, May). High dimensional Bayesian optimization via restricted projection pursuit models. In Artificial Intelligence and Statistics (pp. 884-892). PMLR.
> - [4] Nayebi, A., Munteanu, A., & Poloczek, M. (2019, May). A framework for Bayesian optimization in embedded subspaces. In International Conference on Machine Learning (pp. 4752-4761). PMLR.
> - [5] Malu, M., Dasarathy, G., & Spanias, A. (2021, July). Bayesian optimization in high-dimensional spaces: A brief survey. In 2021 12th International Conference on Information, Intelligence, Systems & Applications (IISA) (pp. 1-8). IEEE.

---

> > ### Author Response · Authors · 2024-11-24
> >
> > Dear Reviewer 4VeV,
> >
> > We hope this message finds you well. We kindly want to check if you had a chance to review our rebuttal, and if you have any further questions or comments we can address to help with your evaluation. Thanks again for your efforts and suggestions in improving our manuscript.
> >
> > Sincerely,
> >
> > The authors

---

### Author Response · Authors · 2024-11-20
**General response to all Reviewers and AC**

We sincerely thank all the reviewers for their insightful and constructive feedback and for dedicating their time to reviewing our work. We appreciate the reviewers for raising their scores and recognizing the significant contribution of our paper in advancing research on new evaluation metrics. We have diligently updated our paper and addressed all comments from each reviewer, and we are eager to respond to any further inquiries or feedback you may have. We ran new experiments and reported in the revision to ensure all concerns are addressed properly.

We are particularly grateful for the acknowledgment of several key aspects of our work:

1. **Alignment with Human Preferences**: Our paper introduces MetaMetrics, which **addresses the critical challenge of aligning automatic evaluation metrics with human preferences in generation tasks**. Given the limitations of current metrics in accurately reflecting human judgments, achieving this alignment is crucial for reducing the reliance on costly and time-consuming human evaluations and improving the generalizability of models across diverse tasks.

2. **Calibrated Weighted Ensemble**: MetaMetrics calibrates existing metrics using Bayesian Optimization and Boosting to create a weighted ensemble. Through comprehensive experiments spanning five different tasks and two modalities (text and image), such as Text Summarization, Machine Translation, Question Answering, Image Captioning, and Reward Model Scoring, we demonstrate its effectiveness.

3. **Versatile and Task-Agnostic Solution**: Our proposed approach, which ensembles existing metrics, **provides a versatile and task-agnostic solution.** We conduct training in low-resource conditions, multi-modality, and evaluate across multiple contexts, including out-of-domain settings.

As per reviewers' requests, we ran new experiments and added the following results:

1. **Enhanced Robustness Section for Out-of-Domain (OOD) Settings (Added New Experiments)**: As requested by a reviewer, we have **expanded the robustness section** in Section F.7, including additional experiments. **Results from cross-lingual and cross-dataset settings are summarized to provide a more comprehensive understanding.**

2. **Ablation Studies (Added New Experiments)**: As requested by a reviewer, we have included **an ablation studies section** in Section F.6. These studies explore different calibration functions (such as Spearman correlation) and GP with various weighting functions (**multiplicative** and **a combination of linear and multiplicative**).

3. **Added more metrics to calibrate (Added New Experiments)**: As requested by a reviewer, we have **added more metrics** to evaluate MetaMetrics for SummEval dataset.

4. **Expanded Baselines (Added New Experiments)**: As requested by a reviewer, we have **added more baselines** for uniform weighting and weighted averaging in Tables 1, 2, 3, and 4.

5. **Added calibration/inference time and cost tables for Summeval (Added New Experiments)**. As requested by a reviewer.

6. **Added spearman evaluation for Summeval (Added New Experiments)**. As requested by a reviewer.

We have added several additional tables to further demonstrate the benefits of MetaMetrics and to address the concerns raised by other reviewers. For your convenience, please note the updated table numbers from our previous discussion:

- Table 12 for MetaMetrics-CAP is now Table 13.
- Tables 16 and 17 for MetaMetrics-MT cross-lingual in WMT-23 and WMT-24 have been renumbered to Table 17 and Table 18, respectively.
- Table 18 for MetaMetrics-CAP Cross-Dataset is now Table 19.

In response to the reviewer H1Lp's request, made just one day before the revision period ended, we will conduct an additional experiment on RLHF for our reward model for camera-ready. This experiment aims to demonstrate that reward models developed using MetaMetrics excel not only in the RewardBench benchmark but also in other LLM benchmarks. While we believe this experiment is not strictly necessary to substantiate the core strengths of our claim, since we have already provided ample evidence across various tasks in the paper, it serves to further illustrate the versatility of MetaMetrics in additional applications. Given the timing of the request and our swift agreement with the reviewer, the results of this experiment will be included in the Appendix of the camera-ready version as a supplementary addition.

**We have answered all concerns from all reviewers** and we hope this work will be useful for future research.

---

> ### Author Response · Authors · 2024-11-23
>
> Dear Reviewers,
>
> We hope this message finds you well. We would greatly appreciate it if you could respond to our rebuttals, particularly to let us know if we have sufficiently addressed their concerns. Thank you very much for your support and understanding.

---

### Meta-Review · Area_Chair_fTa3 · 2024-12-20

**Metareview:**

This paper proposed a metametrics that is more aligned with human preference on evaluating the generation tasks. The idea is quite simple, which can be summarized based on the rebuttal comment:
1. Identify the downstream task, such as image captioning.
2. Train MetaMetrics using datasets that include human ratings. Evaluate the performance on an in-domain held-out set, selecting the MetaMetrics based on the performance of this same in-domain held-out set (not the test set). The test set is not used at this stage.
3. Evaluate the selected MetaMetrics on an unseen test dataset split that includes human ratings. This is a standard machine learning evaluation. We use Kendall or Pearson correlation to assess language model behavior relative to human preferences. The test dataset split can include both in-domain unseen test data and out-of-domain unseen test data (e.g., cross-lingual and cross-dataset scenarios). For the in-domain setting, use 70% of the dataset as the test split. For the out-of-domain setting, we use the whole dataset as test set.

The proposed approach demonstrates the effectiveness and usefulness in evlauation and shows good human alignment.

Strength:
1. The paper studies an important task (evaluation for generation task)
2. The approach is simple yet effective
3. The paper is easy to follow

Weakness:
1. The author might need to follow the reviewer suggestions to provide more ablation study such as evaluations of the LLMs before and after alignment.

Overall, I think this is a good paper and recommend accept.

**Additional Comments On Reviewer Discussion:**

The main debates are rooted in Reviewer nKfe and the author on the OOD setting. The root causes seem the reviewer is not persuaded that such simple combination can achieve good perfomrance on the OOD setting and the reviewer also gets confused about how to apply such approach. During the rebuttal, the author fully addressed the reviewer's concerns. The reviewer replies the rebuttal and answers are satisfied. Although the score is remained the same, I think this paper should be recommended accept.

---

### Decision · Program_Chairs · 2025-01-22

Accept (Poster)